# Dynamic-dLLM: Dynamic Cache-Budget and Adaptive Parallel Decoding for Training-Free Acceleration of Diffusion LLM

**Tianyi Wu**[*1]    **Xiaoxi Sun**[*1]    **Yanhua Jiao**[1]    **Yulin Li**[1]
**Yixin Chen**[2]    **Yunhao Cao**[2]    **Yiqi Hu**[2]    **Zhuotao Tian**[‡1,3]
[1]Harbin Institute of Technology, Shenzhen    [2]Huawei    [3]Shenzhen Loop Area Institute

## Abstract

Diffusion Large Language Models (dLLMs) offer a promising alternative to autoregressive models, excelling in text generation tasks due to their bidirectional attention mechanisms. However, their computational complexity, scaling as $\mathcal{O}(L^3)$ with sequence length $L$, poses significant challenges for long-sequence and real-time applications, primarily due to the lack of compatibility with key-value caching and the non-autoregressive nature of denoising steps. Existing acceleration methods rely on static caching or parallel decoding strategies, which fail to account for the dynamic behavior of token properties across layers and decoding steps. We propose **Dynamic-dLLM**, a training-free framework that enhances dLLM inference efficiency through two components: Dynamic Cache Updating (DCU), which adaptively allocates cache-update budgets based on layer-wise token dynamics, and Adaptive Parallel Decoding (APD), which dynamically calibrates decoding thresholds to balance generation quality and efficiency. Extensive experiments on models like LLaDA-8B-Instruct, LLaDA-1.5, and Dream-v0-7B-Instruct across benchmarks such as MMLU, GSM8K, and HumanEval demonstrate that Dynamic-dLLM significantly improves inference speed, attaining an average speedup of exceeding $3\times$ while maintaining performance. Dynamic-dLLM outperforms state-of-the-art acceleration methods and provides a plug-and-play solution for efficient dLLM deployment without compromising performance. The code is available at https://github.com/TianyiWu233/DYNAMIC-DLLM.

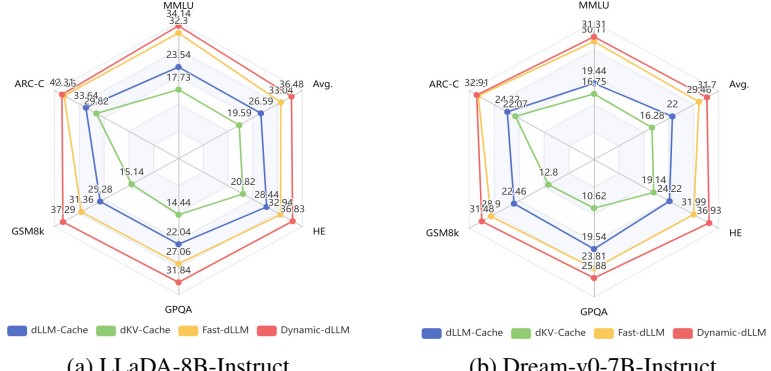

(a) LLaDA-8B-Instruct        (b) Dream-v0-7B-Instruct

Figure 1: The comparison in terms of tokens-per-second (TPS)

## 1 Introduction

Diffusion Large Language Models (dLLMs) have emerged as a compelling alternative to autoregressive models (ARMS), demonstrating strong performance in text generation tasks. Notable examples such as LLaDA (Nie et al., 2025; Zhu et al., 2025) and Dream (Ye et al., 2025) highlight the rapid

---

[*]Equal contribution. The work is done during Tianyi's internship at CBG Celia DeviceAI Team, Huawei.
[‡]Corresponding author (tianzhuotao@hit.edu.cn).

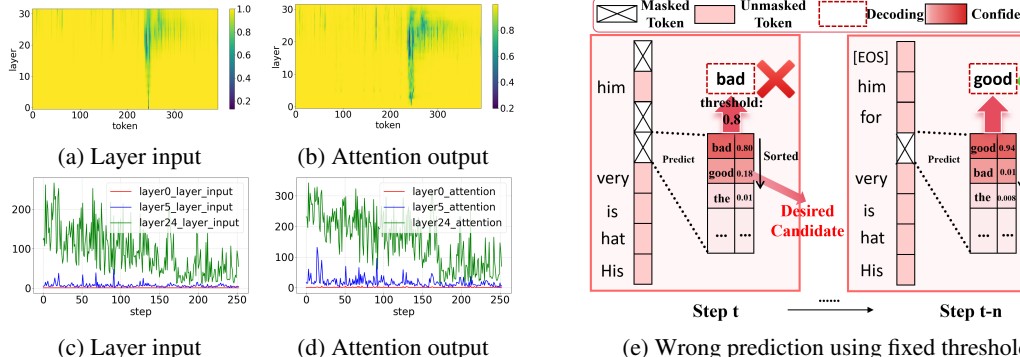

(a) Layer input     (b) Attention output

(c) Layer input     (d) Attention output     (e) Wrong prediction using fixed threshold

Figure 2: (a-b) Layer input similarity and attention output similarity across adjacent denoising steps. The brighter region denotes a higher similarity, indicating most tokens are stable across steps. (c-d) The number of tokens requiring updates across different steps. Differences across layers indicate varying demands for the token update budget. (e) Existing parallel decoding methods may yield wrong predictions as potential candidates have been discarded by the fixed threshold.

progress in this direction. A key advantage of dLLMs lies in their bidirectional attention mechanisms, which enhance scalability and enable superior performance in handling complex scenarios, such as the "reversal curse" (Berglund et al.), where traditional ARMs often struggle. This allows dLLMs to capture richer contextual dependencies in challenging scenarios.

However, despite their strong performance in certain domains, dLLMs face a fundamental challenge: their computational complexity scales as $\mathcal{O}(L^3)$ with respect to sequence length $L$, significantly exceeding the $\mathcal{O}(L^2)$ cost of autoregressive models (ARs). This cubic scaling imposes a severe bottleneck for long-sequence and real-time generation tasks, limiting the practical deployability of dLLMs in latency-sensitive applications. The root cause lies in the non-autoregressive nature of dLLMs, where each denoising step requires updating all tokens in parallel across the full sequence. Besides, this paradigm hinders the caching of key-value activations from previous steps, rendering dLLMs incompatible with the widely used KV-Cache mechanism.

**Key observations.** To address this issue, recent work has explored strategies for dLLM acceleration. For example, (Liu et al., 2025b; Ma et al., 2025; Song et al., 2025) reduce redundancy by caching internal token representations across decoding steps. Concurrently, (Wu et al., 2025) accelerates inference by enabling parallel unmasking of multiple tokens within a single step. These methods implicitly rely on specific token properties, such as feature stability and confidence, to identify opportunities for optimization. However, they all rely on a static strategy across all layers and decoding steps, applying the same caching or unmasking criteria throughout the model and generation process, thus overlooking the dynamic nature of token behavior during generation.

As illustrated in Figure 2(a-d), the token properties vary across different layers and steps. The frequency of changes in the internal features of tokens differs across layers, while the distributions of token confidence fluctuate across decoding steps. The static strategies adopted by existing methods may fail to account for this dynamic behavior, leading to performance degradation. Therefore, this observation prompts a critical question: *how to design an adaptive method that dynamically aligns with the model's intrinsic layer-wise and step-wise token dynamics to improve the efficiency?*

**Our solution.** In this work, we propose **Dynamic-dLLM**, a training-free framework for accelerating dLLM inference. Dynamic-dLLM consists of two key components: Dynamic Cache Updating (DCU) and Adaptive Parallel Decoding (APD).

Specifically, as tokens may exhibit heterogeneous dynamics across layers, instead of a static cache updating strategy across all layers, we propose Dynamic Cache Updating (DCU) that allocates cache-update budgets adaptively, ensuring that layers requiring frequent updates are prioritized, while computational overhead is reduced in stable layers. In addition, the existing parallel decoding strategy with fixed thresholds risks committing to tokens prematurely, as confidence estimates can shift over time, leading to error propagation. To mitigate this, we introduce Adaptive Parallel De-

coding (APD) that dynamically calibrates decoding thresholds by tracking the evolving distribution of prediction confidence, achieving a decent trade-off between the degradation of generation quality caused by a low threshold and the inefficiency resulting from a high threshold.

Extensive experiments across LLaDA-8B-Instruct, LLaDA-1.5, Dream-v0-7B-Instruct, and benchmarks covering mathematics, science, coding, and general tasks demonstrate the effectiveness and strong generalization capabilities of the proposed method. Notably, Dynamic-dLLM achieves a maximum acceleration of up to 4.48×, with an average speedup exceeding 3× while still maintaining performance, making it a plug-and-play training-free solution for enhancing the efficiency of dLLMs without compromising performance. In summary, our contributions are as follows:

- In this study, we observe that the variations across layers and decoding steps of dLLM may undermine the effectiveness of existing static rule-based acceleration methods.
- We propose Dynamic-dLLM, a training-free framework composed of Dynamic Cache Updating (DCU) and Adaptive Parallel Decoding (APD), DCU adaptively allocates cache-update budgets across layers, while APD dynamically calibrates decoding thresholds across steps, jointly enabling efficient yet robust acceleration of dLLMs.
- Extensive experiments across diverse models and tasks show that Dynamic-dLLM substantially improves inference efficiency while preserving the accuracy, outperforming state-of-the-art acceleration methods.

## 2 BACKGROUND AND MOTIVATION

### 2.1 PRELIMINARIES OF DLLM

In this section, we introduce preliminaries regarding the inference process of dLLM (Nie et al., 2025). The introduction of related work is presented in the Appendix D due to the page limit.

Given a prompt of length $L_{\text{prompt}}$ tokens and a target generation length of $L_{\text{gen}}$ tokens, let $L = L_{\text{prompt}} + L_{\text{gen}}$. The dLLM generates the output in $T$ iterative decoding steps, producing approximately $L_{\text{gen}}/T$ tokens per step. Let $\mathcal{V}$ denote the model's vocabulary, and let $[\text{MASK}] \in \mathcal{V}$ be a special placeholder token indicating positions to be predicted. Denote by $\mathbf{x}^t \in \mathcal{V}^L$ the token sequence at step $t$, where $t = T, T-1, \ldots, 0$. The initial sequence is constructed as:

$$\mathbf{x}^T = (x_0, \ldots, x_{L_{\text{prompt}}-1}, [\text{MASK}], \ldots, [\text{MASK}]), \tag{1}$$

where $x_i$ are the given prompt tokens. At each step $t$, the mask predictor $f_\theta$ computes a distribution over the vocabulary for each position:

$$\mathbf{z}^t = f_\theta(\mathbf{x}^t) \in \mathbb{R}^{L \times |\mathcal{V}|}. \tag{2}$$

Using greedy decoding, we can obtain the most probable token at each masked position:

$$\hat{x}_i^t = \arg\max_{v \in \mathcal{V}} \left( \text{Softmax}(\mathbf{z}_i^t) \right)_v, \quad \text{if } x_i^t = [\text{MASK}]. \tag{3}$$

A transition function $S$ then updates the sequence to $\mathbf{x}^{t-1}$ by selectively replacing tokens based on confidence scores, re-masking low-confidence predictions to refine them in subsequent steps: $\mathbf{x}^{t-1} = S(\hat{\mathbf{x}}^t, \mathbf{x}^t, t)$. The final output sequence $\mathbf{x}^0$ is yielded when $t = 0$.

### 2.2 KEY OBSERVATIONS

Despite recent progress in accelerating diffusion-style LLMs (dLLMs) (Liu et al., 2025b; Wu et al., 2025; Ma et al., 2025; Song et al., 2025), two critical inefficiencies remain unaddressed.

**Layer-wise Cache Update Needs Vary Significantly.** Existing methods exploit temporal redundancy by reusing cached intermediate features (*e.g.*, query, key, value, attention output, FFN output) from the previous step for a subset of tokens, assuming high feature similarity across steps. However, as illustrated in Figure 2(a-d), the proportion of tokens requiring cache updates varies substantially across layers, increasing monotonically from shallow to deep layers. This suggests that uniform or heuristic caching strategies are suboptimal. Instead, *a layer-adaptive cache update policy is essential for dynamically allocating computation budgets where they matter most*.

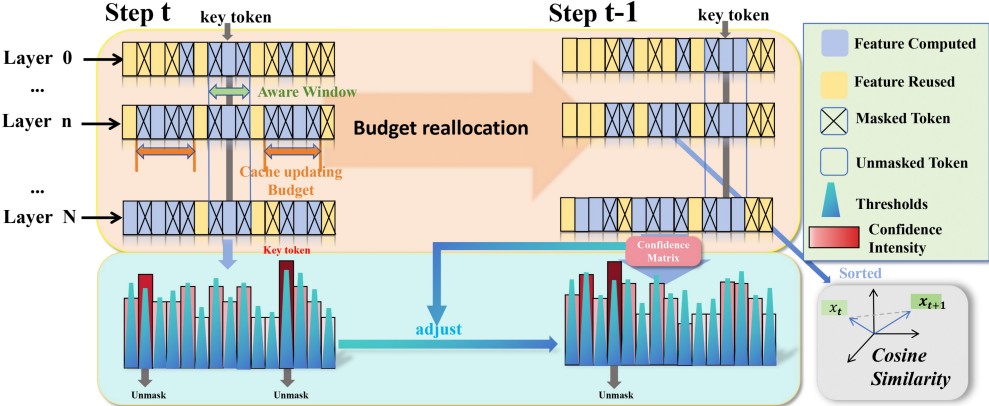

Figure 3: Dynamic-dLLM consists of two key components: Dynamic Cache Updating (DCU, upper part) and Adaptive Parallel Decoding (APD, lower part). DCU reallocates cache update budget for each layer at each step, while APD dynamically adjusts the decoding thresholds for all tokens.

**Static Thresholding Hinders Parallel Decoding Efficacy.** Parallel decoding strategy (*e.g.*, Wu et al. (2025)) unmask tokens once their confidence exceeds a fixed threshold. Yet, as shown in Figure 2(e), the token with the highest confidence at an early step may not be the desired output and will be revised later, often replaced by its "runner-up" prediction with the second-highest confidence initially. Conversely, tokens whose top prediction exhibits clear dominance over alternatives, *i.e.*, low entropy or large margin, can be safely finalized earlier, even if absolute confidence remains below a static threshold. Therefore, to enable earlier commitment to stable predictions, thereby expediting convergence without compromising accuracy, *exploring the feasibility of a dynamic per-token threshold, adjusting adaptively based on the predicted distribution (e.g., entropy or probability margin), becomes essential.*

## 3 METHOD

To overcome the limitations of existing approaches, we propose Dynamic-dLLM, a training-free acceleration framework that dynamically optimizes dLLM inference along two dimensions: cache-update management and parallel decoding scheduling.

Regarding cache-update management, we introduce a dynamic allocation mechanism for managing cache updates, recognizing the varying dynamics across layers. This approach dynamically distributes the update budget among layers, prioritizing layers that require more frequent cache updates. On the other hand, for optimizing the parallel decoding, we replace fixed confidence thresholds with an adaptive per-token unmasking strategy, based on the predicted distribution of each token. This strategy facilitates early commitment to confident predictions while postponing uncertain ones, achieving a more balanced trade-off between speed and output quality.

The overview is presented in Figure 3. Sections 3.1 and 3.2 detail each component, respectively.

### 3.1 DYNAMIC CACHE UPDATING

Recent works (Liu et al., 2025b; Ma et al., 2025; Song et al., 2025) update a fixed or uniform number of token caches across all layers. However, as demonstrated in Section 2.2, the demand for cache updates varies significantly across layers. This observation motivates the need for a dynamic allocation strategy that adapts the cache-update budget per layer according to the specific requirement.

In this section, we propose the Dynamic Cache Updating (DCU) strategy, which selectively updates only those tokens whose representations undergo significant changes between consecutive inference steps. Prior work (Liu et al., 2025b) identifies such tokens by measuring the cosine similarity between the current and cached Value vectors. While effective, this approach incurs non-negligible computational overhead due to the explicit recomputation and comparison of Value vectors. Ideally,

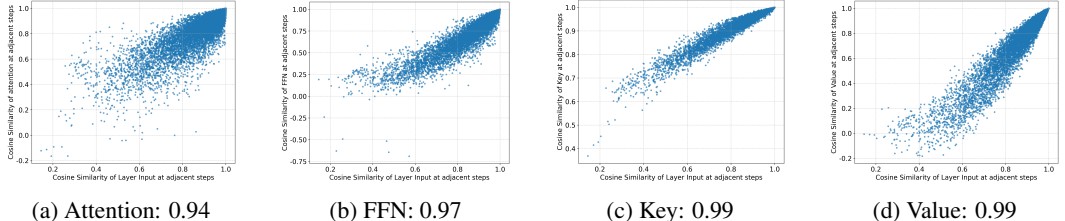

(a) Attention: 0.94      (b) FFN: 0.97      (c) Key: 0.99      (d) Value: 0.99

Figure 4: Spearman correlation values of layer inputs with intermediate features, including Key, Value, Attention Output, and FFN Output. We visualized the cosine similarity between tokens' feature vectors and their cached counterparts at adjacent steps, and compared the relationship between layer input and (a) Attention Output, (b) FFN Output, (c) Key, (d) Value.

if token dynamics could be estimated without recomputing these vectors, cached values could be safely reused, thereby reducing redundancy.

Inspired by Liu et al. (2025a), who observed a strong correlation between model inputs and outputs in diffusion transformers (DiT) (Peebles & Xie, 2023a), we investigate the relationship between layer inputs and intermediate features in dLLMs. As shown in Figure 4, the features cached (*e.g.*, Key, Value, Attention Output, and FFN Output) exhibit high correlation with the corresponding layer inputs. This implies that changes in layer inputs across steps serve as a reliable proxy for the underlying dynamics of intermediate activations. Consequently, input-level differences can effectively inform cache-update decisions without accessing or recomputing the cached features themselves.

**Layer-Adaptive Cache Budget Allocation.** To dynamically allocate the cache update budget across layers, we first define a token-level dissimilarity metric, $d_i^{t,l}$, estimating the change in the representation of token $x_i$ at layer $l$ between consecutive inference steps $t$ and $t + 1$. This metric is computed using the cosine distance between the normalized token inputs at the respective steps:

$$d_i^{t,l} = 1 - \frac{(\mathbf{x}_i^{t,l})^\top \mathbf{x}_i^{t+1,l}}{\|\mathbf{x}_i^{t,l}\|\|\mathbf{x}_i^{t+1,l}\|} \tag{4}$$

A higher value of $d_i^{t,l}$ denotes a greater change in the token's representation, suggesting a higher need for cache update. Then, we aggregate the token-level variations into a layer-wise metric $s^{t,l}$. This metric represents the average change in token representations within layer $l$:

$$s^{t,l} = \frac{1}{N} \sum_{i=0}^{N-1} d_i^{t,l}, \tag{5}$$

where $N$ is the sequence length. Subsequently, the cache update budget for layer $l$ at step $t$, denoted as $B_{\text{layer}}^{t,l}$, is then allocated proportionally to its measured dynamism at the previous step $(t+1)$, $s^{t+1,l}$. This allocation is normalized across all layers using the total available budget, $B_{\text{layer}} \times \texttt{LayerNum}$:

$$B_{\text{layer}}^{t,l} = (B_{\text{layer}} \times \texttt{LayerNum}) \cdot \frac{s^{t+1,l}}{\sum_{k=0}^{\texttt{LayerNum}-1} s^{t+1,k}}. \tag{6}$$

For each layer $l$, the set of tokens scheduled for cache update at step $t$, denoted $\mathcal{U}^{t,l}$, is initialized as an empty set at the start of the step: $\mathcal{U}^{t,l} \leftarrow \emptyset$. Then, layer $l$ identifies the set $\mathcal{S}^{t,l}$ comprising the top-$B_{\text{layer}}^{t,l}$ tokens with the highest variation $d_i^{t,l}$. These selected tokens are then added to the update set: $\mathcal{U}^{t,l} \leftarrow \mathcal{U}^{t,l} \cup \mathcal{S}^{t,l}$.

**Token Stuck in the Mud.** Nevertheless, the layer-adaptive cache budget allocation strategy may potentially make some tokens *stuck in the mud*. Specifically, if a token $x_i$ is not selected for an update in layer $l$, its cached representation remains unchanged. Consequently, its input to layer $l + 1$ also remains static, leading to a zero variation score $d_i^{t,l+1} = 0$ for that layer. As the allocation strategy prioritizes tokens with high $d_i^{t,l+1}$, the token $x_i$ will only be updated in layer $l + 1$ if the number of tokens exhibiting non-zero variation is insufficient to fill the allocated budget $B_{\text{layer}}^{t,l+1}$. Should this

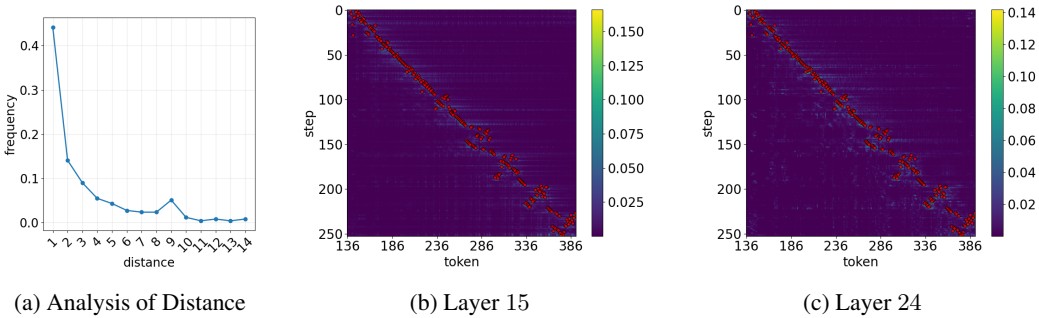

(a) Analysis of Distance          (b) Layer 15          (c) Layer 24

Figure 5: Local property analysis of dLLMs (a) Relationship between the distance from key token and the frequency of being decoded in the current step. The closer the token is to the key token, the higher the probability of it being decoded. (b) The last two images respectively represent the attention of response tokens to key token in layer 15 and 24. Red dot is the key token at this step. The illustration shows that the tokens around the key token have higher attentions, which means that the changes caused by decoding the key token affect those tokens more than others.

occur, and if $x_i$ is again not selected (*e.g.*, chosen randomly among the low-priority tokens), it will remain unchanged entering layer $l+2$, perpetuating the cycle. We refer to this phenomenon, where a token fails to be updated across multiple consecutive layers due to consistently low variation scores induced by prior missed updates, as a token becoming *stuck in the mud*.

**Mandatory Update Window.** As illustrated in Figure 5, there exists a spatial locality in the update pattern: tokens surrounding the one unmasked in the previous step (the *key token*) are statistically more likely to be updated in the current step. Let the position of the key token be $p$. To mitigate the risk of the *next key token* (the token with the highest confidence to be unmasked in the current step) becoming *stuck in the mud*, we introduce a *Mandatory Update Window*. This mechanism ensures that a local region around the key token is always updated, regardless of the adaptive budget allocation. Formally, we define a window of fixed size $B_{\text{window}}$ centered on the key token's position $p$. The set of token positions covered by this window at a given step is $\left[p - \frac{B_{\text{window}}}{2}, p + \frac{B_{\text{window}}}{2}\right]$. For each layer $l$, the caches for all tokens within this window are compulsorily added to the layer's update set $\mathcal{U}^{t,l}$:

$$\mathcal{U}^{t,l} \leftarrow \mathcal{U}^{t,l} \cup \left\{ x_i \,\middle|\, p - \frac{B_{\text{window}}}{2} \leq i \leq p + \frac{B_{\text{window}}}{2} \right\}. \tag{7}$$

This updated set $\mathcal{U}^{t,l}$ then constitutes the final list of tokens whose caches will be recomputed for layer $l$ in the current step. By ensuring continuous updates within this local window, we reduce the likelihood of critical tokens being overlooked and retain the response to local changes. The global budget is subsequently distributed adaptively among the remaining tokens based on the layer-specific variation metrics $s^{t,l}$ for the following step.

## 3.2 ADAPTIVE PARALLEL DECODING

Section 2.2 highlights that the peak confidence of a token can vary significantly across decoding steps in dLLMs. This inherent dynamism poses a challenge for fixed-threshold parallel decoding methods (Wu et al., 2025), which rely on a static criterion and consequently suffer from decoding inaccuracies due to mispredictions at certain steps.

To address this, we introduce the *Adaptive Parallel Decoding* mechanism that dynamically adjusts the masking threshold for each token based on its local prediction stability. Each token $x_i$ starts with an initial threshold $\tau_i^T$. The threshold at step $t$, denoted $\tau_i^t$, is adapted from the threshold used at the previous step $t+1$, $\tau_i^{t+1}$.

**Adaptive Threshold via Confidence Concentration.** The core idea is to modulate the threshold based on the concentration of the token's predicted probability distribution. Intuitively, a diffuse distribution (a small gap between the highest and second-highest probabilities) suggests lower confidence in the current prediction, warranting a stricter (higher) threshold to reduce unnecessary up-

| | MMLU | ARC-C | GSM8k | GPQA | HE | Avg. |
|---|---|---|---|---|---|---|
| **LLaDA-8B-Instruct** | 60.92 | **88.53** | 78.21 | 32.17 | **38.54** | 59.67 |
| Throughput (TPS, ↑) | 10.19(1.0×) | 25.49 (1.0×) | 8.32 (1.0×) | 7.60 (1.0×) | 15.54 (1.0×) | 1.0× |
| + dLLM-Cache | 61.33 | 88.49 | 78.78 | 31.84 | 37.99 | 59.69 |
| Throughput (TPS, ↑) | 23.54 (2.31×) | 33.64 (1.32×) | 25.28 (3.0×) | 22.04( 2.90×) | 28.44(1.83×) | 2.27× |
| + dKV-Cache | 61.37 | 87.98 | **79.17** | 32.25 | 38.23 | **59.80** |
| Throughput (TPS, ↑) | 17.73 ( 1.74×) | 29.82 (1.17×) | 15.14 (1.82×) | 14.44( 1.90×) | 20.82 (1.34×) | 1.59× |
| + Fast-dLLM | **61.39** | 88.05 | 78.44 | 32.01 | 38.21 | 59.62 |
| Throughput (TPS, ↑) | 26.29 (2.58×) | 32.88 (1.29×) | 22.74 (2.73×) | 22.65( 2.98×) | 27.82 (1.79×) | 2.27× |
| + Dynamic-dLLM (Ours) | 60.95 | 88.09 | 78.24 | 31.98 | 38.33 | 59.51 |
| Throughput (TPS, ↑) | **30.16** (2.96×) | 40.27 (1.58×) | 27.21 (3.27×) | 25.46 (3.35×) | 30.61 (1.97×) | 2.63× |
| + Fast-dLLM* | 61.08 | 88.32 | 76.62 | 32.21 | 37.87 | 59.22 |
| Throughput (TPS, ↑) | 32.30 (3.17×) | 41.55 (1.63×) | 31.36 (3.77×) | 27.06(3.56×) | 32.94 (2.12×) | 2.85× |
| + Dynamic-dLLM* (Ours) | 60.89 | 87.79 | 78.01 | 31.89 | **38.08** | 59.33 |
| Throughput (TPS, ↑) | **34.14 (3.35×)** | 42.31 (1.66×) | 37.29 (4.48×) | 31.84(4.19×) | 36.83(2.37×) | 3.21× |

Table 1: Results on LLaDA-8B-Instruct (Nie et al., 2025). Each cell includes the accuracy, decoding throughput (TPS), with relative efficiency enhancement to the baseline. Best values in bold, suboptimal values underlined. Results with * are obtained with parallel decoding.

dates. Conversely, a concentrated distribution indicates stability, permitting a reduced threshold for early decoding. Let $\mathbf{z}_i^t$ be the probability distribution over the vocabulary $\mathcal{V}$ for token $x_i$ at step $t$, the index of the most likely token is: $u = \arg\max_{v \in \mathcal{V}} (\mathbf{z}_i^t)_v$. Thus, the concentration of this distribution is quantified using the second-highest probability score:

$$c_i^t = 1 - \max_{v \in \mathcal{V} \setminus \{u\}} (\mathbf{z}_i^t)_v. \tag{8}$$

A larger value of $c_i^t$ signifies a more peaked and confident distribution. Based on this measure, the decoding threshold for token $x_i$ at step $t$ is adjusted as follows:

$$\tau_i^t = \tau_i^{t+1} - \alpha \cdot c_i^t, \tag{9}$$

where $\alpha$ is a positive hyperparameter controlling the sensitivity of the threshold adaptation. This formulation ensures that tokens with highly concentrated distributions (large $c_i^t$) have their thresholds decreased, allowing for early decoding, while tokens with diffused distributions have increased thresholds to prevent decoding errors.

**Integration with Temporal Instability.** In addition, the magnitude of historical shifts in a token's confidence distribution provides a strong signal for its likelihood of future revision. We quantify this shift via the cosine distance between the token's confidence distributions at adjacent steps:

$$H_i^t = 1 - \frac{(\mathbf{z}_i^t)^\top \mathbf{z}_i^{t+1}}{\|\mathbf{z}_i^t\| \|\mathbf{z}_i^{t+1}\|}. \tag{10}$$

A larger $H_i^t$ indicates greater instability in the prediction, suggesting that the token may still be undergoing refinement and thus warrants a stricter (higher) threshold to prevent early decoding. By combining $c_i^t$ and $H_i^t$, the decoding threshold for token $x_i$ at step $t$ is updated as:

$$\tau_i^t = \tau_i^{t+1} - \alpha \cdot c_i^t + \beta \cdot H_i^t, \tag{11}$$

where $\alpha, \beta \geq 0$ are hyperparameters balancing the influence of prediction confidence and temporal instability.

Algorithms 1 and 2 in the Appendix outline the core mechanisms of Dynamic-dLLM for accelerating dynamic LLMs (dLLMs) via Feature-Caching and Parallel Decoding, respectively. By explicitly accounting for dynamism along both the layer and step dimensions, Dynamic-dLLM minimizes redundant computation and thereby significantly accelerates the inference process of dLLMs.

## 4 EXPERIMENT

### 4.1 EXPERIMENT SETTINGS

We assessed the performance of Dynamic-dLLM using three typical dLLMs as baselines: LLaDA-8B-Instruct, LLaDA-1.5 and Dream-7B-Instruct. If not otherwise specified, we default $B_{\text{layer}}$ to 32, and $B_{\text{window}}$ to 32. More experimental details are shown in Appendix B.

| | MMLU | ARC-C | GSM8k | GPQA | HE | Avg. |
|---|---|---|---|---|---|---|
| **LLaDA-1.5** | 61.42 | **88.51** | 81.62 | 33.61 | **40.24** | **61.08** |
| Throughput (TPS, ↑) | 10.18 (1.0×) | 25.50 (1.0×) | 8.30 (1.0×) | 7.57 (1.0×) | 15.54 (1.0×) | 1.0× |
| + dLLM-Cache | 61.39 | 88.04 | **81.64** | 33.45 | 40.18 | 60.94 |
| Throughput (TPS, ↑) | 23.62 (2.32×) | 33.92 (1.33×) | 26.48 (3.19×) | 21.88 (2.89×) | 28.28 (1.82×) | 2.31× |
| + dKV-Cache | **61.47** | 88.23 | 81.53 | 33.54 | 39.81 | 60.92 |
| Throughput (TPS, ↑) | 17.41 (1.71×) | 29.33 (1.15×) | 15.27 (1.84×) | 14.46 (1.91×) | 20.67 (1.33×) | 1.59× |
| + Fast-dLLM | 61.46 | 88.18 | 81.21 | **33.63** | 40.21 | 60.94 |
| Throughput (TPS, ↑) | 26.47 (2.60×) | 33.15 (1.30×) | 23.07 (2.78×) | 22.56 (2.98×) | 28.13 (1.81×) | 2.29× |
| + Dynamic-dLLM (Ours) | 61.03 | 88.29 | 80.98 | 33.37 | 39.93 | 60.72 |
| Throughput (TPS, ↑) | 30.03 (2.95×) | 41.06 (1.61×) | 27.31 (3.29×) | 25.28 (3.34×) | 30.77 (1.98×) | 2.63× |
| + Fast-dLLM* | 61.22 | 88.31 | 80.94 | 33.43 | 40.06 | 60.79 |
| Throughput (TPS, ↑) | 32.17 (3.16×) | 41.31 (1.62×) | 31.13 (3.75×) | 27.10 (3.58×) | 32.93 (2.12×) | 2.85× |
| + Dynamic-dLLM* (Ours) | 61.34 | 88.02 | 81.03 | 32.97 | 40.01 | 60.67 |
| Throughput (TPS, ↑) | **34.20 (3.36×)** | **42.59 (1.67×)** | **37.02 (4.46×)** | **31.57 (4.17×)** | **36.67 (2.36×)** | **3.20×** |

Table 2: Results on LLaDA-1.5 (Zhu et al., 2025). Each cell includes the accuracy, decoding throughput (TPS), with relative efficiency enhancement to the baseline. Best values in bold, suboptimal values underlined. Results with * are obtained with parallel decoding.

| | MMLU | ARC-C | GSM8k | GPQA | HE | Avg. |
|---|---|---|---|---|---|---|
| **Dream-v0-7B-Instruct** | **73.34** | 89.63 | 77.47 | 34.08 | **56.82** | **66.27** |
| Throughput (TPS, ↑) | 9.97 (1.0×) | 20.44 (1.0×) | 8.05 (1.0×) | 7.13 (1.0×) | 14.95 (1.0×) | 1.0× |
| + dLLM-Cache | 73.08 | 90.04 | 76.64 | **34.75** | 54.31 | 65.76 |
| Throughput (TPS, ↑) | 19.44 (1.95×) | 24.32 (1.19×) | 22.46 (2.79×) | 19.54 (2.74×) | 24.22 (1.62×) | 2.06× |
| + dKV-Cache | 72.93 | 89.40 | 77.32 | 33.87 | 54.69 | 65.64 |
| Throughput (TPS, ↑) | 16.75 (1.68×) | 22.07 (1.08×) | 12.80 (1.59×) | 10.62 (1.49×) | 19.14 (1.28×) | 1.42× |
| + Fast-dLLM | 72.14 | **90.11** | 76.81 | 33.99 | 55.70 | 65.75 |
| Throughput (TPS, ↑) | 18.74 (1.88×) | 29.05 (1.47×) | 21.50 (2.67×) | 17.04 (2.39×) | 20.18 (1.35×) | 1.95× |
| + Dynamic-dLLM (Ours) | 72.09 | 89.25 | 77.28 | 33.17 | 54.93 | 65.34 |
| Throughput (TPS, ↑) | 25.82 (2.59×) | 31.89 (1.56×) | 25.52 (3.17×) | 21.90 (3.07×) | 27.81 (1.86×) | 2.45× |
| + Fast-dLLM* | 71.97 | 89.98 | 76.95 | 33.34 | 56.78 | 65.80 |
| Throughput (TPS, ↑) | 30.11 (3.02)× | 32.50 (1.59×) | 28.90 (3.59×) | 23.81 (3.34×) | 31.99 (2.14×) | 2.74× |
| + Dynamic-dLLM* (Ours) | 72.10 | 89.38 | **77.52** | 32.02 | 54.05 | 65.01 |
| Throughput (TPS, ↑) | **31.31 (3.14×)** | **32.91 (1.61×)** | **31.48 (3.91×)** | **25.88 (3.63×)** | **36.93 (2.47×)** | **2.95×** |

Table 3: Results on Dream-v0-7B-Instruct (Ye et al., 2025). Each cell includes the accuracy, decoding throughput (TPS), with relative efficiency enhancement to the baseline. Best values in bold, suboptimal values underlined. Results with * are obtained with parallel decoding.

To comprehensively evaluate a model's performance and efficiency, we employ two key metrics: accuracy on benchmarks and throughput, with the latter measured in Tokens Per Second (TPS). The benchmarks includes MMLU (5-shot)(Hendrycks et al., 2020), ARC-challenge (ARC-c, 0-shot)(Clark et al., 2018), GPQA (5-shot)(Rein et al., 2024), GSM8k (4-shot)(Cobbe et al., 2021), and HumanEval (HE, 0-shot)(Chen et al., 2021). For fair comparison, we divided the methods into two groups, one using Feature-Cache(Liu et al., 2025b; Ma et al., 2025; Wu et al., 2025) and the other using KV-Cache and parallel decoding(Wu et al., 2025). All experiments were performed on NVIDIA Pro6000 GPUs.

## 4.2 MAIN RESULTS

Our results (baseline vs. alternative methods vs. our Dynamic-dLLM) are presented in Table 1, 2, and 3. These results show that Dynamic-dLLM not only achieves the most significant throughput improvement but also maintains performance.

With only feature cache enabled, Dynamic-dLLM delivers substantial speedups for high-priority tasks without accuracy degradation. It achieves notable throughput boosts on benchmarks with an average speedup of over 2.5× across all evaluated tasks for LLaDA-8B-Instruct , while maintaining accuracy. When combined with parallel decoding, Dynamic-dLLM scales speedups. For LLaDA-8B-Instruct on GSM8k, throughput hits 37.29 TPS (4.48× faster than the baseline's 8.32 TPS), with average speedup across tasks reaching 3.21× and robust accuracy.

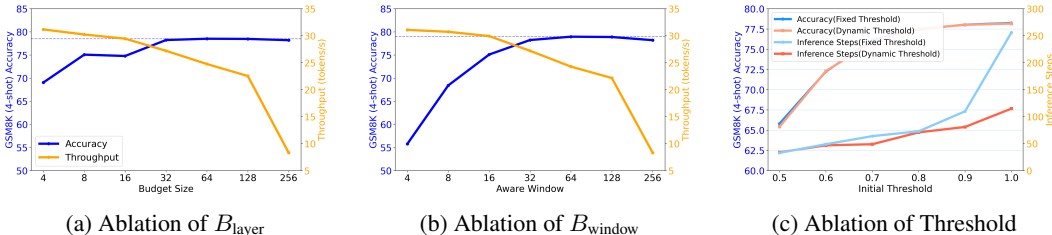

| (a) Ablation of $B_{layer}$ | (b) Ablation of $B_{window}$ | (c) Ablation of Threshold |
|---|---|---|

Figure 6: Ablation studies on key hyperparameters, investigating the respective effects on the model's performance (accuracy) and efficiency (throughput).

This superiority persists across models: LLaDA-1.5 achieves 4.46× speedup on GSM8k (37.02 vs. 8.30 TPS) with near-baseline accuracy (60.67% vs. 61.08%); Dream-v0-7B-Instruct gains 3.91× speedup on GSM8k (31.48 vs. 8.05 TPS). These cross-model results demonstrate its generalization capabilities.

### 4.3 ABLATION STUDIES

In this section, we present the ablation studies regarding the core designs of our method.

**Impact of $B_{layer}$ on Accuracy and Throughput.** As shown in Figure 6a, we fix the $B_{window}$ to 32 and do not use parallel decoding, and explore the impact of $B_{layer}$ on accuracy and throughput. With the gradual increase of $B_{window}$, the accuracy shows an upward trend, reaching a plateau around 32. On the other hand, the throughput also rapidly decreases with the increase of $B_{window}$. Based on observations, a value of 32 for $B_{window}$ is a more trade-off choice.

**Impact of $B_{window}$ on Accuracy and Throughput.** Similarly, we discussed the impact of $B_{window}$ on accuracy and throughput in Figure 6b. $B_{window}$ is fixed to 32 and parallel decoding is disabled. The impact of $B_{window}$ on accuracy and throughput is roughly the same as that of $B_{layer}$, but the smaller $B_{window}$ has a more severe reduction in accuracy than $B_{layer}$. To ensure that the accuracy is basically on par with the baseline, we have chosen 32 as the optimal value for $B_{window}$.

**Dynamic Threshold vs. Fixed Threshold**. We discussed the difference between fixed threshold and dynamic threshold in Figure 6c. The accuracy of both is the same under all initialization. However, dynamic thresholds bring fewer inference steps than fixed thresholds in higher initialization. With the maximum initialization of 0.9, which does not excessively descend performance, dynamic thresholds can reduce inference steps by approximately 30% compared to the fixed thresholds.

## 5 CONCLUDING REMARKS

**Summary.** We present Dynamic-dLLM, a training-free framework for accelerating diffusion LLMs by adapting to the dynamic behavior of tokens across layers and decoding steps. By introducing dynamic cache updating and adaptive parallel decoding, our method significantly reduces redundant computation while preserving generation quality. Extensive experiments across various models and benchmarks covering mathematics, science, coding, and general tasks demonstrate the effectiveness and strong generalization capabilities of the proposed method. In summary, Dynamic-dLLM offers a general, plug-and-play solution toward efficient dLLM inference, highlighting the importance of adaptive strategies in non-autoregressive generation.

**Limitation & Future Work.** While Dynamic-dLLM demonstrates strong performance across standard language generation benchmarks, its capabilities in multi-modal understanding and complex reasoning scenarios remain largely unexplored. In particular, the model's current design is tailored to unimodal textual inputs, and it is unclear how its core mechanisms generalize to settings involving heterogeneous data modalities (e.g., vision, audio, or structured knowledge). Future work should investigate how these principles can be reformulated or extended to address the unique challenges of cross-modal alignment, representation fusion, and modality-specific computational demands. Such extensions could unlock new avenues for building more flexible and efficient foundation models capable of robust reasoning across diverse input types.

ACKNOWLEDGEMENT

This work was supported by the Guangdong Basic and Applied Basic Research Foundation (2025A1515011546) and by the Shenzhen Science and Technology Program (JCYJ20240813105901003, KJZD20240903102901003, ZDCY20250901113000001).

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

CONTENTS

## A  ALGORITHM SUPPLEMENT

The pseudocode of Dynamic-dLLM is shown in Algorithm 1 and 2.

---

**Algorithm 1** Dynamic Cache Updating

---

**Require:** Mask predictor $f_\theta$, prompt $\mathbf{c}$ and initial masked sequence $\mathbf{x}^T$ with length $L$, denoising steps $T$, cache update budget $B_{\text{window}}$ and $B_{\text{layer}}$, initial threshold $\tau^T$.

**Ensure:** Final prediction $\mathbf{x}^0$
$\quad\quad\quad\quad\quad\quad\quad\quad\quad\quad\quad\quad\quad\quad\quad\quad\quad\quad\quad\quad$ ▷ /* Initialize caches at step $t = T$ */
1: $C \leftarrow \text{InitializeCache}(L, \mathbf{x}^T)$ $\quad$ ▷ Cache Key, Value, Attention Output and FFN Output of $L$ tokens.
2: Generate prediction $\hat{\mathbf{x}}^T$ using model $f_\theta$ $\quad\quad\quad\quad$ ▷ Needs initial pass or separate handling
3: $\mathbf{x}^{T-1} \leftarrow \mathcal{S}(\hat{\mathbf{x}}^T, \mathbf{x}^T, T)$
4: **for** $t = T - 1$ down to 1 **do**
5: $\quad$ $\mathbf{x}_{\text{layer\_in}} \leftarrow \mathbf{x}^t$ $\quad\quad\quad\quad\quad\quad\quad\quad\quad\quad\quad\quad$ ▷ Initial input for layer 0 at step $t$
6: $\quad$ **for** $l$ in each layer in the Transformer network **do**
7: $\quad\quad$ $\mathbf{x}^{t,l} \leftarrow \text{LayerNorm}(\mathbf{x}^l_{layer\_in})$
8: $\quad\quad$ $\mathcal{U}^{t,l} \leftarrow \emptyset$
9: $\quad\quad$ $\mathcal{U}^{t,l} \leftarrow \mathcal{U}^{t,l} = \mathcal{U}^{t,l} \cup \left\{ x_i \mid p - \frac{B_{\text{window}}}{2} \leq i \leq p + \frac{B_{\text{window}}}{2} \right\}$ ▷ /* $p$ is the position of token unmasked in last step */
10: $\quad\quad$ **for** each token $j$ in sequence **do**
11: $\quad\quad\quad$ $d^{t,l}_j = 1 - \frac{(\mathbf{x}^{t,l}_j)^\top \mathbf{x}^{t+1,l}_j}{\|\mathbf{x}^{t,l}_j\|\|\mathbf{x}^{t+1,l}_j\|}$
12: $\quad\quad$ **end for**
13: $\quad\quad$ $s^{t,l} \leftarrow \text{Mean}\left(d^{t,l}_0, d^{t,l}_1, \ldots, d^{t,l}_{L-1}\right)$
14: $\quad\quad$ $B^{t,l}_{\text{layer}} \leftarrow (B_{\text{layer}} \times LayerNum) \cdot \frac{s^{t+1,l}}{\sum_{k=0}^{LayerNum-1} s^{t+1,k}}$
15: $\quad\quad$ $\mathcal{U}^{t,l} \leftarrow \mathcal{U}^{t,l} \cup$ indices of $B^{t,l}_{\text{layer}}$ tokens with heightest $d^{t,l}_j$
16: $\quad\quad$ $\mathbf{x}_{\text{layer\_out}}, C \leftarrow \text{RefreshCache}(\mathbf{x}^t_{\text{norm}}, l, C, \mathcal{U}^{t,l})$
17: $\quad\quad$ $\mathbf{x}_{\text{layer\_in}} \leftarrow \mathbf{x}_{\text{layer\_out}}$ $\quad\quad\quad\quad\quad\quad\quad\quad$ ▷ Update input for the next layer
18: $\quad$ **end for** $\quad\quad\quad\quad\quad\quad\quad\quad\quad\quad\quad\quad\quad\quad\quad\quad\quad$ ▷ End layer loop
19: $\quad$ Generate prediction $\mathbf{z}^t$ using final layer output $\mathbf{x}_{\text{layer\_out}}$
20: $\quad$ $\mathbf{x}^{t-1}, \tau^{t-1} \leftarrow \text{ParallelDecoding}(\mathbf{z}^t, \mathbf{x}^t, \tau^t)$ $\quad$ ▷ Adaptive Parallel Decoding shown in 2
21: $\quad$ **if** all $\mathbf{x}^{t-1}$ unmasked **then**
22: $\quad\quad$ **break**
23: $\quad$ **end if**
24: **end for** $\quad\quad\quad\quad\quad\quad\quad\quad\quad\quad\quad\quad\quad\quad\quad\quad\quad\quad\quad\quad$ ▷ End step loop
25: **return** final prediction $\mathbf{x}^0$

---

## B  EXPERIMENT DETAILS

### B.1  BENCHMARKS AND SETTINGS

Table 4 shows the specific setups for each benchmark, involving the count of decoding steps, block length, and generation length. The benchmarks encompass MMLU (5-shot), ARC-C (0-shot), GSM8K (4-shot), Math (4-shot), and HumanEval (0-shot). To examine the generalization and robustness of diverse approaches, we reduce task-dependent hyperparameter adjustments and instead use a uniform setup for all benchmarks except HumanEval. Owing to its unique task characteristic, HumanEval demands a greater number of decoding steps and a longer generation length.

### B.2  IMPLEMENTATION DETAILS

We offer a thorough explanation of the parameter setups for the compared methods dKV-Cache and dLLM-Cache across various models. According to the suggested configurations in the dKV-Cache paper, we set the cache update interval to 8 for the LLaDA series and to 4 for the Dream series.

---

**Algorithm 2** Adaptive Parallel Decoding

---

**Require:** Prediction $\mathbf{z}^t$, parameters $\alpha, \beta$, initial threshold $\tau_n^T$ for every masked token $n$, masked sequence $\mathbf{x}^t$

1: $\mathbf{p}^t \leftarrow \text{Softmax}(\mathbf{z}^t)$          ▷ Probability distribution over $\mathcal{V}$
2: **for** $n = 0$ to $L - 1$ **do**
3:      $\mathbf{p}_{\text{sorted}} \leftarrow \text{Sort}(\mathbf{p}_n^t, \text{descending})$
4:      $c_i^t \leftarrow 1 - \mathbf{p}_{\text{sorted}}[1]$
         ▷ /* Calculate peak concentration */
5: **end for**          ▷ End token loop
         ▷ /* Calculate confidence fluctuation (for adjacent steps) */
6: **if** $t < T - 1$ **then**
7:      **for** $i = 0$ to $L - 1$ **do**
8:          $H_i^t \leftarrow 1 - \frac{(\mathbf{z}_i^t)^\top \mathbf{z}_i^{t+1}}{\|\mathbf{z}_i^t\| \|\mathbf{z}_i^{t+1}\|}$          ▷ Cosine similarity
9:      **end for**
10: **end if**
         ▷ /* Adaptive threshold adjustment */
11: **if** $t < T - 1$ **then**
12:      $\tau_i^t = \tau_i^{t+1} - \alpha \cdot c_i^t + \beta \cdot H_i^t$
13: **end if**
14: Unmask all $i$ with $p_i^t \geq \tau^t$, always unmask $p_i^t$
15: **return** $\mathbf{x}_{t-1}$

---

For dLLM-Cache, the paper presents multiple parameter configurations, where $K_p$ denotes the prompt refresh interval and $K_r$ represents the response refresh interval. specifically, For LLaDA-8B-Instruct: $K_p = 50$, $K_r = 7$; For LLaDA-1.5: $K_p = 100$, $K_r = 6$; For Dream-v0-7B-Instruct: $K_p = 50$, $K_r = 2$.

In addition, for Adaptive Parallel Decoding (APD) in Dynamic-dLLM, we set $\alpha = 0.001$ and $\beta = 0.0008$ based on extensive statistical analysis.

| Datasets | Steps | Block Len | Gen Len |
|----------|-------|-----------|---------|
| MMLU | 256 | 32 | 256 |
| ARC-C | 256 | 32 | 256 |
| GSM8K | 256 | 32 | 256 |
| Math | 256 | 32 | 256 |
| HumanEval | 512 | 32 | 512 |

Table 4: Configuration of Benchmarks

## C EXAMPLE DESCRIPTION

As shown in Figure 7, in the absence of candidate, a fixed threshold can actually hinder early decoding of correct predictions, while Adaptive Parallel Decoding monitors the status of candidates in real time and ends unnecessary steps early.

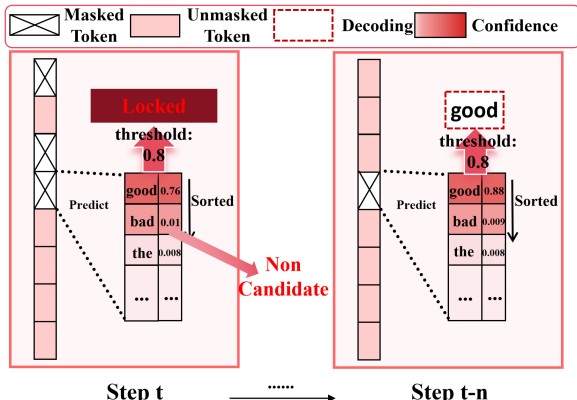

Figure 7: Fixed threshold may hinder the early output of correct predictions, as shown in the figure. The correctly predicted "good" cannot be decoded until its confidence exceeds the threshold $0.8$, which wastes $n$ steps.

# D    RELATED WORK

**Large Language Models.** Driven by transformer-based architectures and large-scale pretraining, Large Language Models (LLMs) have achieved remarkable success, demonstrating exceptional capabilities(Tian et al., 2020; Lai et al., 2021; Jiang et al., 2021; Peng et al., 2023; Cui et al., 2023; Luo et al., 2023; Shao et al., 2024; Peng et al., 2024a). While these models primarily excel in textual processing(Tian et al., 2022; 2023; Ning et al., 2023; Wang et al., 2024), their robust architectural foundation has paved the way for various functional extensions(Cui et al., 2022; 2023; Peng et al., 2024b; Yang et al., 2024; Wang et al., 2024), such as semantic segmentation and object detection(Yang et al., 2024; 2025; Lai et al., 2024b; Peng et al., 2025; Wang et al., 2025b; Huang et al., 2025a). One notable derivative branch is the development of Multimodal Large Language Models (MLLMs)(Wang et al., 2025a; Li et al., 2025; Zhang et al., 2025; Huang et al., 2025b), which expand the core LLM utility by integrating modality encoders to process inputs beyond text, such as image, audio, and video. Through this extension, the reasoning power of the central LLM backbone has been adapted to assist in other domains, including traditional computer vision tasks, as illustrated by specific applications like LISA(Lai et al., 2024a) and LISA++(Yang et al., 2023) in reasoning segmentation.

**Diffusion Large Language Models.** Diffusion models, which excel in continuous data generation through iterative denoising processes (Sohl-Dickstein et al., 2015; Ho et al., 2020), have recently shown promising potential in natural language processing. Unlike their success in image domains (Rombach et al., 2022; Peebles & Xie, 2023b), adapting these models to text generation faces fundamental challenges arising from the discrete token space and sequential dependencies inherent in language. A key advancement in addressing these issues comes from discrete diffusion frameworks, particularly Masked Diffusion Models (MDMs) that operate by progressively refining sequences through context-aware mask prediction (Austin et al., 2021; Lou et al., 2023). Recent methodological innovations have significantly expanded the capabilities of diffusion-based language models. Scaling efforts have produced foundation models like LLaDA(Nie et al., 2025), an 8B parameter bidirectional Transformer trained from scratch, and Dream(Ye et al., 2025) which leverages pretrained autoregressive weights, both achieving performance parity with similarly sized autoregressive models.

**Inference acceleration methods of dLLMs.** Multiple studies have investigated strategies for speeding up discrete diffusion large language models (dLLMs). Some studies using feature caching cut down on computational costs by caching the internal features of tokens across different diffusion steps. dLLM-Cache (Liu et al., 2025b) selects a fixed proportion of tokens for cache update for each layer by sorting the cosine similarity of Value vector between adjacent steps. dKV-Cache (Ma et al., 2025) puts the tokens decoded at each step into the cache and doeses not update them in later steps. Fast-dLLM (Wu et al., 2025) puts tokens outside the current block to the cache and updates tokens

| $B_{window}$ | 32 | 64 | 128 | 192 | 64 | 0 | 256 |
|---|---|---|---|---|---|---|---|
| $B_{layer}$ | 32 | 64 | 128 | 64 | 192 | 256 | 0 |
| score | 73.92 | 77.62 | 79.15 | 78.03 | 78.92 | 74.07 | 75.74 |

Table 5: Performance of DCU with different settings of $B_{window}$ and $B_{layer}$, using 1024 generated tokens on the GSM8K dataset with the LLaDA-8B-Instruct model.

| $gen\_len$ | 256 | 512 | 1024 |
|---|---|---|---|
| score(dynamic dLLM) | 78.01 | 78.31 | 78.15 |
| TPS | 37.29 | 35.41 | 33.19 |
| score(baseline) | 78.21 | 78.98 | 79.11 |
| TPS | 8.78 | 7.62 | 6.21 |

Table 6: Performance scores under different generation lengths with $\alpha = 0.001, \beta = 0.0008$

within the current block. The similarity of these methods is that they adopt the same cache update strategy for all layers, which ignores the different requirements of each layer. In addition, Fast-dLLM proposes the parallel decoding strategy, unmasking the tokens with confidence exceeding a predetermined threshold, which has difficulty balancing quality and efficiency.

# E  EXPERIMENTAL SUPPLEMENTS

## E.1  STABILITY ANALYSIS OF HYPERPARAMETERS

**Stability Analysis of $B_{layer}$ and $B_{window}$.** To investigate the stability of parameter settings across diverse scenarios, we conducted additional experiments. As shown in Table 5, when the sequence length exceeds 1k, the accuracy of the default settings ($B_{layer} = 32$, $B_{window} = 32$) exhibits a slight decline. However, with appropriate increases in these two parameters, the accuracy gradually recovers. Furthermore, when the sum of $B_{layer}$ and $B_{window}$ is fixed, different proportional allocations between them lead to varying impacts on performance. Through these experiments, we confirmed that setting $B_{layer}$ equal to $B_{window}$ yields optimal results. To ensure the method's stability across different generation lengths, we propose an auto-tuning strategy: $B_{layer} = B_{window} = \frac{1}{8} \times gen\_len$ (where $gen\_len$ denotes the generation length).

**Stability Analysis of $\alpha$ and $\beta$.** For the hyperparameters $\alpha = 0.001$ and $\beta = 0.0008$, we performed experiments under different output length settings (256, 512, and 1024), while maintaining $B_{window} = \frac{1}{8} \times gen\_len$ and $B_{layer} = \frac{1}{8} \times gen\_len$ (see Table 6). The results indicate that there is almost no degradation in accuracy across these different generation lengths, verifying the strong stability of $\alpha$ and $\beta$ .

As per Equation 10, $\alpha$ and $\beta$ regulate threshold adaptation to prediction confidence and temporal instability, respectively. We conducted ablation experiments on GSM8K ($gen\_len$=256), with results in Table 7 (score=accuracy; NIS=Number of Inference Steps, fewer = faster). The optimal range for $\alpha/\beta$ is the $10^{-3}$ order of magnitude. The model is sensitive to order-of-magnitude scaling (severe quality loss with over-scaling) but stable to small variations within this range.

## E.2  ANALYSIS OF ADDITIONAL LATENCY

We present comprehensive metrics in Table 8, which details the performance scores (accuracy), average single-inference latency, and additional latency introduced by DCU and APD separately—all evaluated on the GSM8K dataset across different generation lengths ($gen\_len$: 256, 512, 1024). As shown in the Table 8, both DCU and APD introduce minimal additional latency, and this overhead exhibits a clear linear scaling trend with generation length, which only accounts for a small portion of the inference latency, far less than the gain it brings.

| $\alpha$ | $\beta$ | Score | NIS |
|---|---|---|---|
| 0.001 | 0.0008 | 78.01 | 95 |
| 0.01 | 0.008 | 69.75 | 54 |
| 0.1 | 0.08 | 59.76 | 16 |
| 0.001 | 0.0007 | 78.75 | 96 |
| 0.001 | 0.0005 | 78.85 | 95 |
| 0.001 | 0.0017 | 78.93 | 97 |

Table 7: Performance scores and Number of Inference Steps (NIS) under different combinations of $\alpha$ and $\beta$, with 256 generate length on GSM8K

| $gen\_len$ | Baseline | | | DCU | | | APD | | |
|---|---|---|---|---|---|---|---|---|---|
| | Score | Time (s) | Extra Time (s) | Score | Time (s) | Extra Time (s) | Score | Time (s) | Extra Time (s) |
| 256 | 78.21 | 29.16 | None | 78.85 | 9.41 | 1.94 | 78.01 | 6.87 | 0.11 |
| 512 | 78.98 | 67.19 | None | 79.31 | 20.68 | 4.31 | 78.31 | 14.46 | 0.19 |
| 1024 | 79.11 | 164.90 | None | 79.15 | 47.36 | 8.90 | 78.15 | 30.85 | 0.33 |

Table 8: Performance metrics (scores and time-related data) of baseline, DCU, and APD under different generation lengths (gen_len). Time values are rounded to two decimal places.

### E.3 PERFORMANCE IN LOW-RESOURCE HARDWARE

To address low-resource hardware deployment, we tested our method with the LLaDA-8B-Instruct model on the GSM8K and HumanEval datasets using an RTX 4090 GPU. As shown in the Table 9, our method still maintains a strong speed-accuracy trade-off under such cost-constrained settings, demonstrating its practicality for deployment on non-high-end hardware.

| Dataset | LLaDA-8B-Instruct | | Dynamic-dLLM(DCU) | | Dynamic-dLLM | |
|---|---|---|---|---|---|---|
| | Score | TPS | Score | TPS | Score | TPS |
| GSM8K | 78.34 | 5.48 | 77.89 | 14.73 | 77.91 | 29.77 |
| HumanEval | 37.47 | 9.74 | 37.24 | 24.32 | 37.15 | 35.19 |

Table 9: Performance Scores and Inference Speed (TPS) of Different Methods on RTX 4090

