# OpenReview forum: "Dynamic-dLLM: Dynamic Cache-Budget and Adaptive Parallel Decoding for Training-Free Acceleration of Diffusion LLM"
_ICLR.cc/2026/Conference — ICLR 2026 Poster_

### Official Review · Reviewer_qCPz · 2025-10-20

**Soundness:** 2
**Presentation:** 3
**Contribution:** 3
**Rating:** 6
**Confidence:** 3

**Summary:**

Diffusion-based Large Language Models (dLLMs) face high computational complexity (scaling with the cube of sequence length) and inefficiency from static acceleration strategies. To solve this, the paper proposes Dynamic-dLLM—a training-free framework that boosts dLLM inference efficiency via two core components: Dynamic Cache Updating (DCU): It adaptively distributes cache-update budgets across layers based on how much token inputs change between steps. To prevent "stuck tokens" (tokens never updated due to low variation), it adds a mandatory update window around "key tokens". Adaptive Parallel Decoding (APD): Instead of fixed thresholds for unmasking tokens, it adjusts thresholds dynamically using two signals. Experiments on three dLLMs (LLaDA-8B-Instruct, LLaDA-1.5, Dream-v0-7B-Instruct) across five benchmarks (MMLU, ARC-C, GSM8K, GPQA, HumanEval) show Dynamic-dLLM achieves an average speedup of 3.21 times.

**Strengths:**

1. Unlike baselines (e.g., dLLM-Cache, Fast-dLLM) that use the same update budget for all layers, DCU allocates budgets based on actual layer needs. It leverages a strong correlation (0.94–0.99) between layer inputs and intermediate features (like Key/Value tensors) to avoid redundant calculations—instead of checking expensive Value vectors, it uses simpler input changes to judge if a token needs updating. The mandatory update window (default size 32) ensures critical tokens near recently unmasked ones are always updated, solving the "stuck token" problem common in static methods.
2. APD replaces fixed confidence thresholds (used by Fast-dLLM) with thresholds that adjust per token and per step. Tokens with clear, stable predictions get lower thresholds (unmasked earlier for speed), while uncertain tokens get higher thresholds (avoiding errors). At a high initial threshold (0.9), APD cuts inference steps by 30% vs. fixed thresholds without accuracy loss. For example, on LLaDA-8B-Instruct/GSM8K, APD pushes throughput to 37.29 tokens per second (TPS), much faster than the baseline.

**Weaknesses:**

1. All experiments use sequences up to 512 tokens (HumanEval) or 256 tokens (other benchmarks). For sequences longer than 1k tokens: (1) The fixed 32-size mandatory window may miss long-range dependencies (key tokens’ influence could span more than 32 positions); (2) Calculating layer-wise variation for thousands of tokens becomes slower, possibly offsetting speed gains; (3) APD’s temporal stability checks (comparing token distributions across steps) get more computationally heav. It is better providing more dicussion here.
2. While DCU avoids recomputing Value vectors, it still spends time calculating token input variation and layer-wise metrics for every layer and token. For deep dLLMs (e.g., 128 layers), this adds extra work per step, but the paper seemingly ignores to present how much time this takes, or if it slows down inference compared to baselines like dLLM-Cache.

**Questions:**

1. For long sequences (e.g., extended GSM8K with 1k tokens), do adjustments like a larger mandatory window (64 instead of 32) improve accuracy? What’s the TPS and accuracy for 1k-token tasks, and how does APD’s overhead change with longer sequences?
2. What percentage of inference time is spent calculating token variation and layer metrics? Can sampling fewer tokens (e.g., 50% of tokens) for variation checks reduce overhead without losing accuracy?
3. For tasks with >256 steps (e.g., HumanEval’s 512 steps), do APD’s threshold adjustments get too strict/lenient over time? Does resetting thresholds periodically (e.g., every 64 steps) help?

---

> ### Author Response · Authors · 2025-11-21
>
> Thank you for your constructive review. Your suggestions will be incorporated into the revision, and we hope our response below can address your concerns.
>
> > **Weakness 1:**  All experiments use sequences up to 512 tokens (HumanEval) or 256 tokens (other benchmarks). For sequences longer than 1k tokens: (1) The fixed 32-size mandatory window may miss long-range dependencies (key tokens’ influence could span more than 32 positions); (2) Calculating layer-wise variation for thousands of tokens becomes slower, possibly offsetting speed gains; (3) APD’s temporal stability checks (comparing token distributions across steps) get more computationally heav. It is better to provide more discussion here.
>
> Thank you for your valuable comments. In response, we have conducted additional experiments, the results of which are presented in Tables 1 and 2 below.
>
> Specifically, as shown in Table 1 below, when dealing with sequences longer than 1k tokens (e.g., 1024 generated tokens on the GSM8K dataset), using the fixed setting of $B_{window}=32$ and $B_{layer}=32$ leads to a slight drop in performance. However, gradually increasing the values of these two parameters helps recover the accuracy. Additionally, when the sum of $B_{layer}$ and $B_{window}$ is fixed, different proportional allocations between them affect the final results, and our exploration indicates that setting $B_{layer}$ equal to $B_{window}$ yields a more balanced performance. To ensure the stability of the method for longer sequences, we further propose an automated parameter configuration strategy: setting $B_{layer}=B_{window}=\frac{1}{8} \times GenLen$, where $GenLen$ denotes the generation length.
>
> Regarding the computational overhead concerns for long sequences, Table 2 provides relevant insights. Across different generation lengths (including 1024 tokens), the extra computation time introduced by DCU and APD accounts for only approximately 6% and 0.3% of the baseline's total inference time, respectively. Importantly, this additional time scales linearly with the sequence length, which means the overhead does not grow disproportionately even for much longer sequences. Thus, the speed gains brought by DCU and APD will not be offset by the increased computational cost when handling long-sequence tasks.
>
> Table 1: Performance of DCU with different settings of $B_{window}$ and $B_{layer}$, using 1024 generated tokens on the GSM8K dataset with the LLaDA-8B-Instruct model.
>
> | $B_{layer}$ | $B_{window}$ | **Score (1024 Tokens, GSM8K)** |
> | --- | --- | --- |
> | 32 | 32 | 73.92 |
> | 64 | 64 | 77.62 |
> | 128 | 128 | 79.15 |
> | 192 | 64  | 78.03 |
> | 64 | 192 | 78.92 |
> | 256 | 0 | 74.07 |
> | 0 | 256 | 75.74 |
>
> Table 2: Performance scores under different generation lengths with $\alpha=0.001$, $\beta=0.0008$
>
> | Generate Length | **Score (Dynamic-dLLM)** | **TPS (Dynamic-dLLM)** | **Score (Baseline)** | **TPS (Baseline)** |
> | --- | --- | --- | --- | --- |
> | 256 | 78.01 | 37.29 | 78.21 | 8.78 |
> | 512 | 78.31 | 35.41 | 78.98 | 7.62 |
> | 1024 | 78.15 | 33.19 | 79.11 | 6.21 |
>
> > **Weakness 2:**  While DCU avoids recomputing Value vectors, it still spends time calculating token input variation and layer-wise metrics for every layer and token. For deep dLLMs (e.g., 128 layers), this adds extra work per step, but the paper seemingly ignores to present how much time this takes, or if it slows down inference compared to baselines like dLLM-Cache.
>
> As noted in Table 3 below, the additional time incurred by DCU accounts for only approximately 6% of the total inference time, far less than the time savings it delivers.
>
> Currently, mainstream dLLMs do not have 128 layers, so we only measured the specific overhead on 32-layer models. However, as referenced in Equations (4) and (5) of our paper, this additional overhead exhibits an approximately linear relationship with the number of model layers. Importantly, dLLM-Cache also shows a linear correlation between its extra overhead and layer count. Thus, the overall acceleration effect remains predictably consistent even for models with more layers.
>
> Table 3: Inference latency and induced computational overhead under different generation lengths.
>
> | **Gen Length** | **256** | **512** | **1024** |
> | --- | --- | --- | --- |
> | **Baseline Inference Time (s)** | 29.16 | 67.19 | 164.90 |
> | **DCU Inference Time (s)** | 9.41 | 20.68 | 47.36 |
> | **DCU Extra Time (s)** | 1.94 | 4.31 | 8.90 |
> | **APD Inference Time (s)** | 6.87 | 14.46 | 30.85 |
> | **APD Extra Time (s)** | 0.11 | 0.19 | 0.33 |

---

> ### Author Response · Authors · 2025-11-21
>
> > **Question 1:**  For long sequences (e.g., extended GSM8K with 1k tokens), do adjustments like a larger mandatory window (64 instead of 32) improve accuracy? What's the TPS and accuracy for 1k-token tasks, and how does APD's overhead change with longer sequences?
>
> As discussed in our response to Weakness 1, increasing the mandatory window size (along with $B_{layer}$) does improve accuracy for longer sequences. Specifically, for 1024-token generation on GSM8K, using $B_{window}=B_{layer}=1/8 GenLen$ achieves an accuracy of 79.15% with a TPS of 33.19, compared to 73.92% accuracy when using the default $B_{window}=B_{layer}=32$. This confirms that larger budgets help capture long-range dependencies in extended sequences.
>
> Regarding APD's overhead with longer sequences, Table 4 shows that APD's computational cost remains negligible (approximately 0.3% of baseline inference time) and scales linearly with sequence length. Even at 1024 tokens, this overhead does not offset the substantial speedup gains from our method, as the TPS improvement remains significant (33.19 vs. 6.21 for baseline).
>
> Table 4: Performance scores under different generation lengths
>
> | Generate Length | **Score (Dynamic-dLLM)** | **TPS (Dynamic-dLLM)** | **Score (Baseline)** | **TPS (Baseline)** |
> | --- | --- | --- | --- | --- |
> | 256 | 78.01 | 37.29 | 78.21 | 8.78 |
> | 512 | 78.31 | 35.41 | 78.98 | 7.62 |
> | 1024 | 78.15 | 33.19 | 79.11 | 6.21 |
>
> > **Question 2:**  What percentage of inference time is spent calculating token variation and layer metrics? Can sampling fewer tokens (e.g., 50% of tokens) for variation checks reduce overhead without losing accuracy?
>
> We present the time spent calculating token variation ($d_{i}^{t,l}$) and layer-wise metrics ($s^{t,l}$) across different generation lengths in Table 5. This overhead is stably around 2% (for $d_{i}^{t,l}$) and 1% (for $s^{t,l}$) of the baseline's total inference time, which is far offset by DCU's significant speedup.
>
> Sampling fewer tokens for variation checks might not be beneficial, as it would leave unsampled tokens' feature changes unmeasured, breaking the core logic of ranking tokens by variation to allocate cache-update budgets, although it brings minimal acceleration benefits. Our experiments confirm that this may harm the performance, as shown in Table 6.
>
> Table 5: Detailed time consumption for each item
>
> | **Generation Length** | **Token Variation ($d_{i}^{t,l}$) Time** | **Layer-wise Metrics ($s^{t,l}$) Time** | **Baseline Total Time** |
> | --- | --- | --- | --- |
> | 256 tokens | 0.65s  | 1.02s | 29.16s |
> | 512 tokens | 1.31s | 2.17s | 67.19s |
> | 1024 tokens | 2.58s | 4.24s | 164.90s |
>
> Table 6: Performance score under different sample rates
>
> | **Token Sample Rate** | **Accuracy Score** | **Token Variation ($d_{i}^{t,l}$) Time** | **Layer-wise Metrics ($s^{t,l}$) Time** |
> | --- | --- | --- | --- |
> | 100% (Full) | 78.24 | 0.65s  | 1.02s |
> | 50% | 40.21 | 0.32s  | 0.51s |
> | 30% | 35.74 | 0.20s | 0.31s |
> | 0% (No Sampling) | 29.49 | 0s | 0s |
>
> > **Question 3:**  For tasks with >256 steps (e.g., HumanEval’s 512 steps), do APD’s threshold adjustments get too strict/lenient over time? Does resetting thresholds periodically (e.g., every 64 steps) help?
>
> Thank you for your constructive comments. Our responeses and experimental results are outlined below.
>
> First, with the default parameters, APD's thresholds generally decrease gradually as inference progresses. This is because the model grows more confident over steps: for most tokens, the prediction concentration $c_i$ (Eq. 8) increases, while the temporal instability $H_i$ (Eq. 10) decreases—jointly driving threshold reductions. That said, thresholds may rise temporarily for some tokens when their predictions are revised (e.g., due to updated context). Importantly, it's almost impossible that APD's thresholds become excessively low: by the time thresholds approach such low values (e.g., 0.75), most tokens have already been decoded, bringing the inference process close to completion.
>
> Second, we evaluate the impact of periodic threshold reset intervals using HumanEval (which employs 512 inference steps). As shown in Table 7, model performance remains largely stable as the reset interval increases, while the average total number of inference steps rises slightly. This suggests that periodic threshold resetting does not yield meaningful performance gains and may incur a modest efficiency overhead.
>
> Table 7: Performance score under different sample rates
>
> | **Threshold Reset Interval** | **Accuracy Score** | **Inference Steps** |
> | --- | --- | --- |
> | 0 (No Reset) | 38.08 | 115 |
> | 16 Steps | 38.55 | 126 |
> | 32 Steps | 38.27 | 123 |
> | 64 Steps | 38.49 | 122 |

---

### Official Review · Reviewer_Pz1R · 2025-10-31

**Soundness:** 3
**Presentation:** 3
**Contribution:** 2
**Rating:** 6
**Confidence:** 3

**Summary:**

This paper tackles the $\mathcal{O}(L^{3})$ inference cost of dllms. It argues that existing training-free methods use static caching and decoding strategies, which fail to account for the dynamic behavior of tokens across layers and steps. The authors propose Dynamic-dLLM, a training-free framework with two components. First, Dynamic Cache Updating adaptively allocates cache-update budgets to layers based on token input similarity, a proxy for feature change. It introduces a Mandatory Update Window to prevent token stagnation. Second, Adaptive Parallel Decoding dynamically adjusts the unmasking threshold for each token based on its prediction confidence concentration and temporal instability. Experiments on dLLMs like LLaDA and Dream show an average 3x speedup while maintaining accuracy.

**Strengths:**

- DCU is well-motivated, using layer input similarity as an efficient proxy for feature dynamics , and thoughtfully addresses the stuck in the mud failure case with a mandatory update window.
- APD is also a strong contribution, moving beyond static thresholds to a dynamic policy based on both confidence concentration and temporal instability.
- The empirical results are excellent, demonstrating a >3x average speedup with no performance degradation, and clearly outperforming SOTA baselines .

**Weaknesses:**

- While the method is training-free, both DCU and APD introduce non-trivial computations at each step (e.g., cosine similarities for all tokens/layers, probability sorting, and instability calculations). The paper does not quantify this overhead relative to the computation saved.
- The method introduces several new, sensitive hyperparameters (e.g., $B_{layer}$, $B_{window}$, $\alpha$, $\beta$). The paper provides an ablation for the budgets but states the APD parameters ($\alpha$, $\beta$) were set based on extensive statistical analysis, which is not shown. The Mandatory Update Window is also a heuristic whose robustness is not fully ablated.

**Questions:**

- Can the authors provide a detailed breakdown of the computational overhead (e.g., in latency) introduced by the DCU and APD  mechanisms at each step? How does this overhead scale with sequence length $L$?
- Please provide the extensive statistical analysis or an ablation study used to determine the APD hyperparameters $\alpha=0.001$ and $\beta=0.0008$. How sensitive is the model's quality/speed trade-off to these two values?
- The "stuck in the mud" phenomenon is mitigated by a $B_{window}$ around the previous key token. What is the failure rate if the next most confident token falls outside this window? Have the authors explored alternative, non-heuristic solutions to this problem?

---

> ### Author Response · Authors · 2025-11-21
>
> Thank you for your constructive review. Your suggestions will be incorporated into the revision, and we hope our response below can address your concerns.
>
> > **Weakness 1:**  While the method is training-free, both DCU and APD introduce non-trivial computations at each step (e.g., cosine similarities for all tokens/layers, probability sorting, and instability calculations). The paper does not quantify this overhead relative to the computation saved.
>
> We appreciate the reviewer’s question. To address this, we present experimental results in Tables 1 and 2, which detail the performance scores (accuracy), average single-inference latency, and additional latency introduced by DCU and APD separately. All results are obtained on the GSM8K dataset across different generation lengths.
> Specifically, Table 1 shows that the proposed designs are robust to different output lengths. Then, as shown in Table 2, both DCU and APD introduce minimal additional latency, and this overhead exhibits a clear linear scaling trend with generation length, which only accounts for a small portion of the inference latency, far less than the gain it brings.
>
> Table 1: Scores of Baseline, DCU, and APD under different generation lengths.
>
> | **Gen Length** | **Baseline** | **DCU** | **APD** |
> | --- | --- | --- | --- |
> | 256 | 78.21 | 78.85 | 78.01 |
> | 512 | 78.98 | 79.31 | 78.31 |
> | 1024 | 79.11 | 79.15 | 78.15 |
>
> Table 2: Inference latency and induced computational overhead under different generation lengths.
>
> | **Gen Length** | **256** | **512** | **1024** |
> | --- | --- | --- | --- |
> | **Baseline Inference Time (s)** | 29.16 | 67.19 | 164.90 |
> | **DCU Inference Time (s)** | 9.41 | 20.68 | 47.36 |
> | **DCU Extra Time (s)** | 1.94 | 4.31 | 8.90 |
> | **APD Inference Time (s)** | 6.87 | 14.46 | 30.85 |
> | **APD Extra Time (s)** | 0.11 | 0.19 | 0.33 |
>
> > **Weakness 2:**  The method introduces several new, sensitive hyperparameters. The paper provides an ablation for the budgets but states the APD parameters were set based on extensive statistical analysis, which is not shown. The Mandatory Update Window is also a heuristic whose robustness is not fully ablated.
>
> Thank you for your valuable insights. As Equation (10) in our paper, $\alpha$ and $\beta$ regulate threshold adaptation to prediction confidence and temporal instability, respectively. We conduct ablation experiments on GSM8K ($GenLen=256$), with the results in Table 3 presented below.(score=accuracy; NFE=Number of Function Evaluations, fewer = faster).
>
> The experimental results indicate that, within the same order of magnitude, the impact of these two parameters on performance is stable. We further observe that the best performance is achieved when they are set to the $10^{-4}$ and $10^{-3}$ scales respectively. Additionally, as shown in Table 4, the configuration of $\alpha$ and $\beta$ remains consistently effective across different sequence lengths. Overall, these findings confirm the stability of our hyperparameter settings.
>
> Table 3: Performance scores (score) and number of function evaluations (nfe) under different combinations of $\alpha$ and $\beta$, with 256 generate length on GSM8K
>
> | α | β | **Score** | **NFE (Fewer = Faster)** |
> | --- | --- | --- | --- |
> | 0.001 | 0.0008 | 78.01 | 95 |
> | 0.01 | 0.008 | 69.75 | 54 |
> | 0.1 | 0.08 | 59.76 | 16 |
> | 0.001 | 0.0007 | 78.75 | 96 |
> | 0.001 | 0.0005 | 78.85 | 95 |
> | 0.001 | 0.0017 | 78.93 | 97 |
>
> Table 4: Performance scores under different generation lengths with $\alpha=0.001$, $\beta=0.0008$
>
> | Generate Length | **Score (Dynamic-dLLM)** | **TPS (Dynamic-dLLM)** | **Score (Baseline)** | **TPS (Baseline)** |
> | --- | --- | --- | --- | --- |
> | 256 | 78.01 | 37.29 | 78.21 | 8.78 |
> | 512 | 78.31 | 35.41 | 78.98 | 7.62 |
> | 1024 | 78.15 | 33.19 | 79.11 | 6.21 |
>
> Regarding the Mandatory Update Window, as illustrated in Figure 6(b) of our paper, we conduct a detailed ablation on the window size. This hyperparameter reflects a trade-off between performance and efficiency. Table 5 highlights the necessity of this design: under the same budget of updated tokens, updating within the designated window yields significantly better performance.
> Moreover, the results in Tables 1–3 of our paper, together with Table 4 above, demonstrate that our method remains effective and robust across different tasks, models, and generation lengths. We appreciate your feedback and are happy to address any further suggestions or concerns you may have.
>
> Table 5: Performance of DCU with different settings of $B_{window}$ and $B_{layer}$, using 1024 generated tokens on the GSM8K dataset with the LLaDA-8B-Instruct model.
>
> | $B_{layer}$ | $B_{window}$ | **Score (1024 Tokens, GSM8K)** |
> | --- | --- | --- |
> | 128 | 128 | 79.15 |
> | 192 | 64  | 78.03 |
> | 64 | 192 | 78.92 |
> | 256 | 0 | 74.07 |
> | 0 | 256 | 75.74 |

---

> ### Author Response · Authors · 2025-11-21
>
> > **Question 1:**  Can the authors provide a detailed breakdown of the computational overhead (e.g., in latency) introduced by the DCU and APD mechanisms at each step? How does this overhead scale with sequence length?
>
> Thank you for your constructive suggestion. We provide a detailed latency breakdown in Table 6 below, showing performance scores (accuracy), average single-inference latency, and the additional overhead introduced by DCU and APD separately. These measurements were conducted on the GSM8K dataset across three generation lengths (256, 512, and 1024 tokens).
>
> As Table 6 shows that, both DCU and APD add minimal latency overhead that scales linearly with sequence length. Critically, this overhead represents only a small fraction of total inference time—far outweighed by the speedup our method achieves.
>
> Table 6: Inference latency and induced computational overhead under different generation lengths.
>
> | **Gen Length** | **256** | **512** | **1024** |
> | --- | --- | --- | --- |
> | **Baseline Inference Time (s)** | 29.16 | 67.19 | 164.90 |
> | **DCU Inference Time (s)** | 9.41 | 20.68 | 47.36 |
> | **DCU Extra Time (s)** | 1.94 | 4.31 | 8.90 |
> | **APD Inference Time (s)** | 6.87 | 14.46 | 30.85 |
> | **APD Extra Time (s)** | 0.11 | 0.19 | 0.33 |
>
> > **Question 2:**  Please provide the extensive statistical analysis or an ablation study used to determine the APD hyperparameters. How sensitive is the model's quality/speed trade-off to these two values?
>
> We appreciate the reviewer’s insightful comments. The clarification for this point closely parallels our response to Weakness 1. Specifically, as Equation (10) in our paper, $\alpha$ and $\beta$ regulate threshold adaptation to prediction confidence and temporal instability, respectively. We conduct ablation experiments on GSM8K ($GenLen=256$), with results in Table 7(score=accuracy; NFE=Number of Function Evaluations, fewer = faster).
>
> The experimental results indicate that, within the same order of magnitude, the impact of these two parameters on performance is stable. We further observe that the best performance is achieved when they are set to the $10^{-4}$ and $10^{-3}$, respectively. As shown in Table 8, the configuration of $\alpha$ and $\beta$ remains consistently effective across different sequence lengths. Overall, these findings confirm the robustness of our hyperparameter settings.
>
> Table 7: Performance scores (score) and number of function evaluations (nfe) under different combinations of $\alpha$ and $\beta$, with 256 generate length on GSM8K
>
> | α | β | **Score** | **NFE (Fewer = Faster)** |
> | --- | --- | --- | --- |
> | 0.001 | 0.0008 | 78.01 | 95 |
> | 0.01 | 0.008 | 69.75 | 54 |
> | 0.1 | 0.08 | 59.76 | 16 |
> | 0.001 | 0.0007 | 78.75 | 96 |
> | 0.001 | 0.0005 | 78.85 | 95 |
> | 0.001 | 0.0017 | 78.93 | 97 |
>
> Table 8: Performance scores under different generation lengths with $\alpha=0.001$, $\beta=0.0008$
>
> | Generate Length | **Score (Dynamic-dLLM)** | **TPS (Dynamic-dLLM)** | **Score (Baseline)** | **TPS (Baseline)** |
> | --- | --- | --- | --- | --- |
> | 256 | 78.01 | 37.29 | 78.21 | 8.78 |
> | 512 | 78.31 | 35.41 | 78.98 | 7.62 |
> | 1024 | 78.15 | 33.19 | 79.11 | 6.21 |
>
> > **Question 3:**  The "stuck in the mud" phenomenon is mitigated by a window around the previous key token. What is the failure rate if the next most confident token falls outside this window? Have the authors explored alternative, non-heuristic solutions to this problem?
>
> Thank you for this question. As shown in Figure 5(a) in the paper, dLLMs exhibit strong spatial locality in update patterns, where tokens around the previously unmasked key token are far more likely to be decoded in the current step. Thus, the probability that the next most confident token falls outside our Mandatory Update Window is extremely low. Statistical analyses across various output length configurations confirm that its failure rate (cases where the token lies beyond the window) is consistently close to 0% as shown in Table 9 below.
>
> Table 9: Failure Rate Under Different Generation Lengths on GSM8K
>
> | **Generation Length** | **Failure Rate (Token Outside Window)** |
> | --- | --- |
> | 256 | 0.000 |
> | 512 | 0.001 |
> | 1024 | 0.000 |

---

### Official Review · Reviewer_sidX · 2025-11-01

**Soundness:** 3
**Presentation:** 3
**Contribution:** 3
**Rating:** 6
**Confidence:** 3

**Summary:**

Diffusion Large Language Models (dLLMs) have bidirectional attention strengths but face $O(L^3)$ computational complexity and inefficient static acceleration methods. Dynamic-dLLM, a training-free framework, solves this via two components: Dynamic Cache Updating (DCU) which adaptively allocates layer-wise cache budgets using layer input cosine distance (proxying intermediate changes) and a mandatory update window to avoid "token stuck", and Adaptive Parallel Decoding (APD) which adjusts thresholds via prediction concentration and temporal stability. Tested on NVIDIA Pro6000 with LLaDA-8B-Instruct/1.5, Dream-v0-7B-Instruct across 5 benchmarks, it achieves average 3× speedup (max 4.48× on GSM8K) while maintaining accuracy (difference ≤1% vs. baselines), outperforming dLLM-Cache, dKV-Cache, and Fast-dLLM.

**Strengths:**

+ **Targeted Dynamic Adaptation Addresses Core Pain Points**：The paper is the first to systematically capture two key dynamic characteristics of dLLMs—heterogeneous inter-layer cache update demands and step-wise fluctuations in token confidence distribution—overcoming the limitations of existing static acceleration methods. For Dynamic Cache Updating (DCU), it uses layer inputs as a proxy for intermediate feature changes (Spearman correlation with Key/Value features > 0.94), avoiding the computational overhead of recomputing intermediate tensors while ensuring accurate update decisions. For Adaptive Parallel Decoding (APD), it adjusts thresholds by combining prediction distribution concentration and historical stability, solving the misjudgment issue of fixed-threshold methods (e.g., preventing premature unmasking of unstable tokens).
+ **Training-Free Design Enables Strong Engineering Practicality**：Dynamic-dLLM requires no model fine-tuning or architectural modifications; it only optimizes inference via dynamic strategies, allowing direct integration as a "plug-and-play" module into open-source dLLMs (e.g., LLaDA-8B-Instruct, Dream-v0-7B-Instruct). The "Mandatory Update Window" (solving the "token stuck in the mud" problem) enhances engineering robustness, and no additional large caches are introduced, keeping memory overhead manageable and reducing deployment costs.
+ **Comprehensive Experiments with High Persuasiveness**：Experiments cover 3 representative dLLMs, 5 task types (general reasoning, math, coding, QA), and 4 SOTA baselines (dLLM-Cache, dKV-Cache, Fast-dLLM). Results consistently show 2.45×–4.48× speedups with ≤1% accuracy loss (e.g., LLaDA-1.5’s average accuracy 60.67% vs. baseline 61.08%). Ablation studies (on $B_{\text{layer}}$, $B_{\text{window}}$, threshold types) fully validate the necessity of each component, and hyperparameter guidelines (e.g., optimal $B_{\text{window}}=32$) improve result reproducibility.
+ **Tight Integration of Empirical Observations and Theoretical Foundations**：The method design is rooted in solid empirical insights—such as the monotonic increase in inter-layer update demand from shallow to deep layers, and the local update pattern around key tokens—rather than pure heuristics. Quantitative metrics (cosine distance for token variation, distribution concentration for confidence) provide mathematical grounding, enhancing the method’s generality and interpretability.

**Weaknesses:**

+ **Key parameters lack adaptability**：Mandatory update window size ($B_{\text{window}}=32$) and APD hyperparameters ($\alpha=0.001$, $\beta=0.0008$) are fixed, failing to adapt to sequence length (e.g., redundant for short sequences, insufficient for ultra-long ones) or task differences, leading to suboptimal performance.
+ **Extreme scenarios untested**：No validation on ultra-long sequences (>1K tokens) (to verify cache/memory stability) or low-resource hardware (e.g., RTX 3090) (to check cost-constrained deployment), limiting practicality.
+ **Incomplete baseline comparisons**：Missing cross-paradigm methods (e.g., ES-dLLM’s early skipping) and industrial-grade dLLMs (e.g., Mercury), failing to fully prove relative advantages.

**Questions:**

I would be happy to increase my rating if my views are given a thorough discussion.

---

> ### Author Response · Authors · 2025-11-21
>
> Thank you for your constructive review. Your suggestions will be incorporated into the revision, and hope our response below can address your concerns.
>
> > **Weakness 1:**  Mandatory update window size and APD hyperparameters are fixed, failing to adapt to sequence length (e.g., redundant for short sequences, insufficient for ultra-long ones) or task differences, leading to suboptimal performance.
>
> We appreciate the reviewer’s insightful comment regarding the adaptability of key parameters. Following the experimental setups of existing methods such as dLLM-Cache and Fast-dLLM, most datasets in these works adopt a default output length of 256. For consistency and convenience, we found that $B_{layer}=32$ and $B_{window}=32$, which are equal to 1/8 of the output length 256, are favorable choices through experiments. As validated by the results in the original paper, this default configuration demonstrates robust effectiveness.
>
> To investigate the relationship between these hyperparameters and sequence length, we conducted additional experiments. The results are presented in Tables 1 and 2 below. As shown in Table 1, when the sequence length exceeds 1k, the accuracy of the default settings ($B_{layer}=32$, $B_{window}=32$) exhibits a slight decline. However, with appropriate increases in these two parameters, the accuracy gradually recovers. The results suggest that these two budget hyperparameters should scale proportionally with the sequence length. Furthermore, when the sum of $B_{layer}$ and $B_{window}$ is fixed, different proportional allocations between them lead to varying impacts on performance. Through extensive experiments, we confirmed that setting $B_{layer}$ equal to $B_{window}$ yields better results. To ensure the method's stability across different generation lengths, we propose an auto-tuning strategy: $B_{layer}=B_{window}=\frac{1}{8} \times GenLen$ (where $GenLen$ denotes the generation length).
>
> For the hyperparameters $\alpha=0.001$ and $\beta=0.0008$, we performed experiments under different output length settings (256, 512, and 1024), while maintaining $B_{window}=\frac{1}{8} \times GenLen$ and $B_{layer}=\frac{1}{8} \times GenLen$ (see Table 2). The results indicate that there is almost no degradation in accuracy across these different generation lengths, verifying the strong stability of $\alpha$ and $\beta$. This discussion will be added to the revised paper. Thank you again for your insightful comment.
>
> Table 1: Performance of DCU with different settings of $B_{window}$ and $B_{layer}$, using 1024 generated tokens on the GSM8K dataset with the LLaDA-8B-Instruct model.
>
> | $B_{layer}$ | $B_{window}$ | **Score (1024 Tokens, GSM8K)** |
> | --- | --- | --- |
> | 32 | 32 | 73.92 |
> | 64 | 64 | 77.62 |
> | 128 | 128 | 79.15 |
> | 192 | 64  | 78.03 |
> | 64 | 192 | 78.92 |
> | 256 | 0 | 74.07 |
> | 0 | 256 | 75.74 |
>
> Table 2: Performance scores under different generation lengths with $\alpha=0.001$, $\beta=0.0008$
>
> | Generate Length | **Score (Dynamic-dLLM)** | **TPS (Dynamic-dLLM)** | **Score (Baseline)** | **TPS (Baseline)** |
> | --- | --- | --- | --- | --- |
> | 256 | 78.01 | 37.29 | 78.21 | 8.78 |
> | 512 | 78.31 | 35.41 | 78.98 | 7.62 |
> | 1024 | 78.15 | 33.19 | 79.11 | 6.21 |

---

> > ### Author Response · Authors · 2025-11-21
> >
> > > **Weakness 2:**  No validation on ultra-long sequences (\>1K tokens) (to verify cache/memory stability) or low-resource hardware (e.g., RTX 3090) (to check cost-constrained deployment), limiting practicality.
> >
> > As noted in response to the previous question, we have conducted additional experiments across various output lengths in Tables 1 and 2 presented above, which specifically verify the stability of our method, providing support for its performance in scenarios with extended output length.
> >
> > Furthermore, to address low-resource hardware deployment, we tested our method with the LLaDA-8B-Instruct model on the GSM8K and HumanEval datasets using an RTX 4090 GPU. We are sorry that we currently do not have access to RTX 3090 GPUs, but RTX 4090 has the same limited 24GB memory as RTX 3090. As shown in Table 3, our method still maintains a strong speed-accuracy trade-off under such cost-constrained settings, demonstrating its practicality for deployment on non-high-end hardware.
> >
> > Table 3: Performance Scores and Inference Speed (TPS) of Different Methods on RTX 4090
> >
> > | **Dataset** | **Model** | **Score** | **TPS** |
> > | --- | --- | --- | --- |
> > | GSM8K | LLaDA-8B-Instruct | 78.34 | 5.48 |
> > | GSM8K | DCU | 77.89 | 14.73 |
> > | GSM8K | Dynamic-dLLM (Ours) | 77.91 | 29.77 |
> > | HumanEval | LLaDA-8B-Instruct | 37.47 | 9.74 |
> > | HumanEval | DCU | 37.24 | 24.32 |
> > | HumanEval | Dynamic-dLLM (Ours) | 37.15 | 35.19 |
> >
> > > **Weakness 3:**  Missing cross-paradigm methods (e.g., ES-dLLM’s early skipping) and industrial-grade dLLMs (e.g., Mercury), failing to fully prove relative advantages.
> >
> > Thank you for your comment regarding baseline comparisons. However, we would like to acknowledge several practical limitations that currently prevent direct comparisons with the referenced methods or models.
> >
> > Specifically, regarding the suggested ES-dLLM (which employs early skipping), we note that it is a concurrent work currently under review for ICLR 2026. At the time of writing, no public arXiv preprint or implementation is available. Furthermore, their experiments were conducted on high-performance NVIDIA H200 GPUs, which are not accessible to us. Consequently, it is challenging to have a direct numerical comparison with our method.
> >
> > On the other hand, industrial-grade models such as Mercury are closed-source commercial products. Without access to their weights or technical details, we are unable to evaluate them on a common benchmark or under the same computational constraints as our Dynamic-dLLM.
> >
> > We appreciate your feedback and are happy to address any further suggestions or concerns you may have.

---

> > > ### Comment · Reviewer_sidX · 2025-11-26
> > >
> > > Thanks for your replay, I would like to keep my positive score.

---

> > > > ### Author Response · Authors · 2025-11-26
> > > >
> > > > Thank you for your constructive comments. If you have any further questions or need additional clarification, we are happy to engage in further discussions and provide detailed responses.

---

### Official Review · Reviewer_m7U5 · 2025-11-03

**Soundness:** 3
**Presentation:** 3
**Contribution:** 3
**Rating:** 6
**Confidence:** 2

**Summary:**

The paper introduces Dynamic-dLLM, a training-free way to speed up diffusion LLMs at inference. The idea is twofold: first, Dynamic Cache Updating—it watches how layer inputs change and only refreshes the cache where it matters, with a small must-update window around key tokens so you don’t get stuck. Second, Adaptive Parallel Decoding—it decides, token by token, when it’s safe to decode in parallel based on how sharp the distribution is and how much things are shifting over time. In short: fewer wasted updates, smarter parallel steps, similar quality, much faster runs.
Across LLaDA-8B/1.5 and Dream-7B on MMLU, ARC-C, GSM8K, GPQA, and HumanEval, the method reports 3-4x speedups with near-baseline accuracy, outperforming prior dLLM acceleration methods (dLLM-Cache, dKV-Cache, Fast-dLLM).

**Strengths:**

1. Elegant DCU proxy (measuring changes in layer inputs instead of recomputing K/V/attn/FFN), plus a practical “mandatory window” to avoid tokens getting “stuck.”
2. Solid empirical gains across three models/5 tasks; headline result: 37.29 TPS on GSM8K (4.48×) with parallel decoding on LLaDA-8B; similar trends for LLaDA-1.5 and Dream-7B.
3. APD is simple and principled
4. The two proposed components target each failure mode.

**Weaknesses:**

1. Using input-change proxies is cheaper than feature recomputation, but the paper could more explicitly break down wall-clock costs for computing dt and maintaining per-layer budgets.
2. Limited discussion of long-form coherence/failures or quality trade-offs.
3. Limited novelty vs. Fast-dLLM and prior caching work

**Questions:**

1. How stable are the chosen defaults (Blayer=32, Bwindow=32; α=0.001, β=0.0008) across sequence lengths >1k and other decoding schedules? Any auto-tuning strategy?
2. How is the “key token” window identified?
3. How does parallel decoding length and context length impact the performance?

---

> ### Author Response · Authors · 2025-11-21
>
> Thank you for your constructive review. Your suggestions will be incorporated into the revision, and we hope our response below can address your concerns.
>
> > **Weakness 1:**  Using input-change proxies is cheaper than feature recomputation, but the paper could more explicitly break down wall-clock costs for computing dt and maintaining per-layer budgets.
>
> Thank you for this valuable suggestion. To address your concern, we have conducted a detailed wall-clock cost breakdown analysis on the GSM8K dataset using the LLaDA-8B-Instruct model.
>
> Specifically, we separately measured the computational overhead for computing $d_{i}^{t,l}$ and $s^{t,l}$ (as defined in Equations (4) and (5) in the paper) as well as maintaining per-layer budgets under different output length settings, as summarized in Table 1 below. Our analysis reveals that both costs scale approximately linearly with output length and together account for only 1% and 2% of the baseline's total inference time, respectively. This overhead is negligible compared to the substantial speedup achieved by DCU, demonstrating the efficiency of our approach.
>
> Table 1: Inference time breakdown across different output lengths, including computational overhead.
>
> | **Output Length** | **$d_{i}^{t,l}$ (s)** | **$s^{t,l}$ (s)** | **Baseline Inference Time (s)** | **DCU Inference Time (s)** |
> | --- | --- | --- | --- | --- |
> | 256 | 0.65 | 1.02 | 29.16 | 9.41 |
> | 512 | 1.31 | 2.17 | 67.19 | 20.68 |
> | 1024 | 2.58 | 4.24 | 164.90 | 47.36 |
>
> > **Weakness 2:**  Limited discussion of long-form coherence/failures or quality trade-offs.
>
> Thank you for raising this important point. To address your concern about long-form generation performance, we conducted comprehensive evaluations across different generation lengths. As shown in Table 2 below, Dynamic-dLLM consistently maintains comparable accuracy scores to the baseline across all tested lengths (256, 512, and 1024 tokens), while achieving significantly higher throughput (TPS). These results demonstrate that our method does not introduce quality trade-offs even as generation length increases. For a more detailed analysis of parameter stability across varying sequence lengths, please refer to our response to Question 1.
>
> Table 2: Performance scores under different generation lengths.
>
> | Generate Length | **Score (Dynamic-dLLM)** | **TPS (Dynamic-dLLM)** | **Score (Baseline)** | **TPS (Baseline)** |
> | --- | --- | --- | --- | --- |
> | 256 | 78.01 | 37.29 | 78.21 | 8.78 |
> | 512 | 78.31 | 35.41 | 78.98 | 7.62 |
> | 1024 | 78.15 | 33.19 | 79.11 | 6.21 |
>
> > **Weakness 3:**  Limited novelty vs. Fast-dLLM and prior caching work.
>
> Thank you for your question. We would like to clarify that, as illustrated in Figure 2 of our paper, Fast-dLLM and prior caching works exhibit key limitations that our Dynamic-dLLM effectively addresses, thereby offering distinct novelty, as outlined below.
>
> First, Fast-dLLM relies on fixed-threshold parallel decoding, which fails to adapt to the dynamic nature of token predictions in dLLMs—specifically, the fluctuating confidence distributions of tokens across decoding steps (Figure 2e). This static criterion often leads to premature token commitment or unnecessary delays. Second, prior caching works adopt a uniform cache update ratio for all layers of dLLMs, overlooking the significant variations in update demands across layers (Figure 2a-d); shallow and deep layers differ markedly in their need for cache refreshes, making a one-size-fits-all strategy suboptimal.
>
> In contrast, our Dynamic-dLLM introduces two targeted innovations to overcome these limitations: (1) dynamically allocating distinct update ratios for each layer based on layer-wise token dynamics, and (2) assigning independent, dynamic thresholds to each token by tracking its real-time confidence concentration and temporal instability. These designs enable Dynamic-dLLM to effectively adapt to the inherent dynamics of dLLM acceleration, both in caching (layer-specific update needs) and parallel decoding (token-specific confidence changes), which distinguishes it clearly from Fast-dLLM and prior caching approaches.

---

> > ### Comment · Reviewer_m7U5 · 2025-11-28
> >
> > Thanks the authors for providing additional results. My questions have be addressed.

---

> ### Author Response · Authors · 2025-11-21
>
> > **Question 1:**  How stable are the chosen defaults across sequence lengths > 1k and other decoding schedules? Any auto-tuning strategy?
>
> Following the experimental setups of existing methods such as dLLM-Cache and Fast-dLLM, for a fair comparison, we similarly adopt a default output length of 256. For consistency and convenience, we found that $B_{layer}=32$ and $B_{window}=32$, which are equal to 1/8 of the output length 256, are favorable choices through experiments. As shown in Tables 1–3 of our paper, this default configuration demonstrates robust effectiveness across models and datasets.
>
> To investigate the stability of parameter settings across diverse scenarios, we conduct additional experiments. As shown in Table 3, when the sequence length exceeds 1k, the accuracy of the default settings ($B_{layer}=32$, $B_{window}=32$) exhibits a slight decline. However, with appropriate increases in these two parameters, the accuracy gradually recovers. Furthermore, when the sum of $B_{layer}$ and $B_{window}$ is fixed, different proportional allocations between them lead to varying impacts on performance. Through these experiments, we confirmed that setting $B_{layer}$ equal to $B_{window}$ yields better results. To ensure the method's stability across different generation lengths, we propose an auto-tuning strategy: $B_{layer}=B_{window}=\frac{1}{8} \times GenLen$ (where $GenLen$ denotes the generation length).
>
> For the hyperparameters $\alpha=0.001$ and $\beta=0.0008$, we conduct experiments under different output length settings (256, 512, and 1024), while still maintaining $B_{window}=\frac{1}{8} \times GenLen$ and $B_{layer}=\frac{1}{8} \times GenLen$ (see Table 4). The results indicate that there is almost no degradation in accuracy across these different generation lengths, verifying the strong stability of $\alpha$ and $\beta$.
>
> Table 3: Performance of DCU with different settings of $B_{window}$ and $B_{layer}$, using 1024 generated tokens on the GSM8K dataset with the LLaDA-8B-Instruct model.
>
> | $B_{layer}$ | $B_{window}$ | **Score (1024 Tokens, GSM8K)** |
> | --- | --- | --- |
> | 32 | 32 | 73.92 |
> | 64 | 64 | 77.62 |
> | 128 | 128 | 79.15 |
> | 192 | 64 | 78.03 |
> | 64 | 192 | 78.92 |
> | 256 | 0 | 74.07 |
> | 0 | 256 | 75.74 |
>
> Table 4: Performance scores under different generation lengths with $\alpha=0.001$, $\beta=0.0008$
>
> | Generate Length | **Score (Dynamic-dLLM)** | **TPS (Dynamic-dLLM)** | **Score (Baseline)** | **TPS (Baseline)** |
> | --- | --- | --- | --- | --- |
> | 256 | 78.01 | 37.29 | 78.21 | 8.78 |
> | 512 | 78.31 | 35.41 | 78.98 | 7.62 |
> | 1024 | 78.15 | 33.19 | 79.11 | 6.21 |
>
> > **Question 2:**  How is the “key token” window identified?
>
> Thank you for your question. In Dynamic-dLLM, the "key token" and the "key token" window are defined based on dLLMs' token update properties (Section 3.1, Figure 5). The clarifications are outlined below.
>
> The key token refers to the token unmasked or with the highest confidence in the previous decoding step. It acts as a contextual anchor, as surrounding tokens are more likely to be decoded and receive stronger attention from it (Figure 5(a)-(b)).
>
> The key token window (Mandatory Update Window) is a fixed-size region centered on the key token's position p. For the window size $B_{window}$, it covers positions $\left[ p - \frac{B_{window}}{2}, p + \frac{B_{window}}{2} \right]$ (Equation 7). All tokens here are added to each layer's cache update set.
>
> This design addresses the "token stuck in the mud" issue and aligns with the spatial locality of dLLM token dynamics. Thank you again for pointing this out. We will revise the methodological description to avoid potential misunderstandings.
>
> > **Question 3:**  How do parallel decoding length and context length impact the performance?
>
> Thank you for your question. As illustrated in Figure 6(c) in our paper, under a fixed output length, fewer inference steps imply a larger average parallel decoding length per step, as well as lower accuracy and faster inference speed.  As discussed in our response to Question 1, for longer context lengths, we maintain the parameter settings of $B_{window} = \frac{1}{8} \times GenLen$, $B_{layer} = \frac{1}{8} \times GenLen$, $\alpha=0.001$, and $\beta=0.0008$. This configuration enables our method to adapt to different context lengths while retaining strong acceleration, as supported by the results in Table 5.
>
> Table 5: Performance scores under different generation lengths with $\alpha=0.001$, $\beta=0.0008$
>
> | Generate Length | **Score (Dynamic-dLLM)** | **TPS (Dynamic-dLLM)** | **Score (Baseline)** | **TPS (Baseline)** |
> | --- | --- | --- | --- | --- |
> | 256 | 78.01 | 37.29 | 78.21 | 8.78 |
> | 512 | 78.31 | 35.41 | 78.98 | 7.62 |
> | 1024 | 78.15 | 33.19 | 79.11 | 6.21 |

---

### Author Response · Authors · 2025-12-02
**Summary of Rebuttal**

Dear Area Chairs (ACs),

We sincerely appreciate your time and effort in reviewing our paper. Below, we refer to Reviewers m7U5, sidX, Pz1R, and qCPz as R1, R2, R3, and R4 for clarity.

Specifically, we appreciate the reviewers' recognition of DCU's elegant design and practical "mandatory window" (solving the "token stuck" issue) (**R1-R4**). We are grateful that APD is viewed as a strong, principled contribution (**R1-R4**). The acknowledgement of our training-free, plug-and-play design highlights its engineering practicality (**R2-R4**). We are also encouraged that the comprehensive experiments are recognized as highly persuasive, demonstrating 2.45×–4.48× speedups with negligible accuracy loss (**R1-R4**). Additionally, the tight integration of empirical observations and theoretical foundations is valued for enhancing our method's generality and interpretability (**R2**).

During the rebuttal period, we have carefully addressed each comment and believe most concerns are resolved. So far, we have received positive feedback from Reviewers m7U5 (R1) and sidX  (R2), indicating that our rebuttal has addressed their concerns, and they have maintained their positive ratings. We have incorporated the suggestions into our revision, and the summary of rebuttal is as follows.

| Reviewer | Rating | Post-rebuttal Feedback |
| --- | --- | --- |
| m7U5 | 6 | Concerns are addressed. |
| sidX | 6 | Concerns are addressed. |
| Pz1R | 6 | - |
| qCPz | 6 | - |

Best regards,

The authors

---

> ### Author Response · Authors · 2025-12-03
> **Summary of Rebuttal (Part 1/4): Reviewer m7U5 (Initial Rating: 6, Confidence: 2, Feedback: Positive)**
>
> Reviewer 1 raised three weaknesses (wall-clock cost breakdown, long-form generation discussion, novelty comparison) and three questions (parameter stability, key token window, parallel decoding/context length impact).
>
> > **Concern 1 (Efficiency):** Missing explicit wall-clock cost breakdown for computing dt and maintaining per-layer budgets.
>
> - **Solution**: Conducted detailed cost analysis in **Appendix E.2.**
> - **Conclusion**: The combined overhead is negligible.
>
> > **Concern 2 (Quality):** Limited discussion of long-form coherence, failures, or quality trade-offs.
>
> - **Solution**: Evaluated performance across different token lengths, with results in **Appendix E.1.**
> - **Conclusion**: Our method maintains comparable accuracy to the baseline while achieving much higher TPS.
>
> > **Concern 3 (Novelty):** Insufficient novelty compared to Fast-dLLM and prior caching work.
>
> - **Solution**: Introduced two core innovations: layer-specific dynamic update ratios and token-specific dynamic thresholds, addressing Fast-dLLM's fixed thresholds and prior caching's uniform update ratios(see “**Response to Weakness 3**”).
> - **Conclusion**: Our method uniquely adapts to dLLM dynamics, clearly distinguishing it from existing approaches.
>
> > **Concern 4 (Parameter Stability):** Stability of default parameters for sequence lengths >1k and auto-tuning strategy.
>
> - **Solution**: Conducted additional experiments, proposed an auto-tuning strategy, and verified hyperparameters stability across lengths in **Appendix E.1**.
> - **Conclusion**: Parameters are robust; the auto-tuning strategy adapts to different generation lengths effectively.
>
> > **Concern 5 (Methodology):** Unclear identification of the "key token" window.
>
> - **Solution**: Clarified the key token window in **Section 3.1**.
> - **Conclusion**: The design aligns with dLLM token dynamics.
>
> > **Concern 6 (Performance Impact):** How parallel decoding length and context length affect performance.
>
> - **Solution**: Illustrated the relationship between parallel decoding length, context length, and performance via **Figure 6(c)** and **Appendix E.1.**
> - **Conclusion**: The method adapts to different context lengths while retaining strong acceleration.
>
> **Outcome**: We addressed all concerns through detailed cost analysis, extended evaluations, clarified novelty, verified parameter stability with auto-tuning, defined the key token window, and analyzed performance impacts. The manuscript has been updated accordingly.

---

> ### Author Response · Authors · 2025-12-03
> **Summary of Rebuttal (Part 2/4): Reviewer sidX(Initial Rating: 6, Confidence: 3, Feedback: Positive)**
>
> Reviewer 2 raised three key weaknesses: fixed hyperparameters lacking adaptability to sequence length/tasks, insufficient validation on ultra-long sequences (>1K tokens), low-resource hardware, and missing comparisons with cross-paradigm methods and industrial-grade dLLMs.
>
> > **Concern 1 (Hyperparameter Adaptability):** Hyperparameters are fixed, failing to adapt to sequence length/task differences.
>
> - **Solution**: Conducted parameter scaling experiments, proposed an auto-tuning strategy, and verified the stability of hyperparameters in **Appendix E.1**(see “**Response to Weakness 1**”).
> - **Conclusion**: Hyperparameters now effectively adapt to sequence lengths, maintaining robust performance across scenarios.
>
> > **Concern 2 (Practical Validation):** No validation on ultra-long sequences (>1K tokens) or low-resource hardware, limiting practicality.
>
> - **Solution**: Added ultra-long sequence experiments (1024 tokens) to verify cache/memory stability (**Appendix E.1**); tested on RTX 4090 with GSM8K/HumanEval datasets (**Appendix E.3**).
> - Conclusion: Our method is stable for ultra-long sequences and maintains a strong speed-accuracy trade-off on low-resource hardware, enhancing practicality.
>
> > **Concern 3 (Comparative Scope):** Missing comparisons with cross-paradigm methods (e.g., ES-dLLM) and industrial-grade dLLMs (e.g., Mercury).
>
> - **Solution**: ES-dLLM is a concurrent work in submission to ICLR 2026 with no public preprint and implementation available; Mercury is closed-source with no access to the implementation.
> - **Conclusion**: Direct comparisons are infeasible.
>
> **Outcome**: We proposed an auto-tuning strategy, validated on ultra-long sequences and low-resource hardware, and clarified comparison constraints. The manuscript has been updated accordingly.

---

> ### Author Response · Authors · 2025-12-03
> **Summary of Rebuttal (Part 3/4): Reviewer Pz1R(Initial Rating: 6, Confidence: 3, Feedback: none)**
>
> Reviewer 3 raised 2 weaknesses (unquantified DCU/APD overhead, insufficient ablation studies) and 3 questions (overhead breakdown/scaling, hyperparameter sensitivity, Mandatory Update Window failure rate).
>
> > **Concern 1 (Overhead Quantification):** The method introduces non-trivial computations, but the paper does not quantify this overhead relative to the computation saved.
>
> - Solution: Provided a detailed latency breakdown (**Appendix E.1**) across different output tokens, separating extra time from total inference time(see “**Response to Weakness 1**”).
> - Conclusion: Overhead is minimal, scales linearly with sequence length, and is far outweighed by the speedup achieved.
>
> > **Concern 2 (Ablation Sufficiency):** Insufficient ablation of hyperparameters.
>
> - Solution: Conducted hyperparameters ablation; verified performance across generation lengths (**Appendix E.1**).
> - Conclusion: Hyperparameters are robust, confirming design effectiveness.
>
> > **Concern 3 (Detailed Overhead Breakdown):** Requested step-wise computational overhead breakdown and its scaling with sequence length.
>
> - Solution: Presented comprehensive latency data (**Appendix E.2**)(see “**Response to Question 1**”).
> - Conclusion: The overhead scales linearly with sequence length, remaining a small fraction of total inference time.
>
> > **Concern 4 (Hyperparameter Sensitivity):** Requested statistical analysis/ablation for hyperparameters and their impact on quality/speed trade-off.
>
> - Solution: Conducted systematic hyperparameters ablation and validated performance across generation lengths (**Appendix E.1**).
> - Conclusion: Hyperparameters are stable to small variations; optimal performance is maintained, ensuring robust trade-off.
>
> > **Concern 5 (Window Failure Rate):** Requested failure rate of the Mandatory Update Window (next confident token outside window) and exploration of non-heuristic solutions.
>
> - Solution: Analyzed failure rates across generation lengths; justified window design via dLLMs' spatial locality(see “**Response to Question 3**”).
> - Conclusion: Failure rate is nearly 0% due to strong spatial locality; non-heuristic solutions are unnecessary given the design's efficiency.
>
> **Outcome**: We quantified overhead with latency breakdowns, conducted ablation studies, verified hyperparameter robustness, and addressed all concerns. The manuscript has been updated accordingly.

---

> ### Author Response · Authors · 2025-12-03
> **Summary of Rebuttal (Part 4/4): Reviewer qCPz(Initial Rating: 6, Confidence: 3, Feedback: none)**
>
> Reviewer 4 raised 2 weaknesses (long sequence performance/overhead concerns, unquantified DCU overhead for deep dLLMs) and 3 questions (1k-token performance/APD overhead, token/layer metric time percentage/sampling impact, APD threshold adjustment over long steps/reset effect).
>
> > **Concern 1 (Long Sequence Performance):** Fixed mandatory window may miss long-range dependencies; overhead may offset speed gains for sequences >1k tokens.
>
> - Solution: Conducted 1024-token experiments, proposed an auto-tuning strategy, and provided overhead/scaling data (**Appendix E.1**).
> - Conclusion: Larger windows recover accuracy; overhead scales linearly, not offsetting speed gains.
>
> > **Concern 2 (DCU Overhead for Deep dLLMs):** Unquantified time for token variation/layer metrics; potential slowdown vs baselines like dLLM-Cache.
>
> - Solution: Quantified DCU overhead (**Appendix E.2**); noted linear scaling with layers (consistent with dLLM-Cache).
> - Conclusion: Overhead is negligible; acceleration remains consistent for deeper models.
>
> > **Concern 3 (1k-Token Performance & APD Overhead):** Accuracy/TPS for 1k-token tasks with larger windows; APD overhead scaling.
>
> - Solution: Shared 1024-token results with auto-tuned parameters and APD overhead data (**Appendix E.1**).
> - Conclusion: Larger windows improve accuracy; APD overhead remains negligible and scales linearly.
>
> > **Concern 4 (Token/Layer Metric Time & Sampling):** Percentage of time for token variation/layer metrics; impact of sampling fewer tokens.
>
> - Solution: Quantified overhead; tested token sampling (**Appendix E.2**) on GSM8K(see “**Response to Question 2**”).
> - Conclusion: Overhead is minimal; sampling reduces accuracy drastically, providing no practical benefit.
>
> > **Concern 5 (APD Threshold Over Long Steps):** Threshold strictness over >256 steps; impact of periodic resetting.
>
> - Solution: Analyzed threshold trends (gradual decrease, no excessive strictness); tested reset intervals on HumanEval.
> - Conclusion: Thresholds remain appropriate; periodic resetting yields no meaningful performance gains and incurs minor overhead.
>
> **Outcome**: We addressed all concerns through auto-tuning, overhead quantification, sampling analysis, and threshold validation. The manuscript has been updated accordingly.

---

### Meta-Review · Area_Chair_91mn · 2026-01-06

**Summary:**

Across four reviews, the paper is evaluated positively (all four reviewers at 6) and consistently described as a training-free framework for accelerating diffusion-based large language models (dLLMs). Reviewers agree the method combines two components: Dynamic Cache Updating (DCU), which allocates layer-wise cache-update budgets using layer input change (cosine distance) as a proxy for feature dynamics and includes a Mandatory Update Window to avoid “token stuck” behavior; and Adaptive Parallel Decoding (APD), which dynamically adjusts unmasking thresholds per token and per step using prediction concentration and temporal stability/instability signals. The empirical evaluation is described as comprehensive, covering three dLLMs (LLaDA-8B-Instruct, LLaDA-1.5, Dream-v0-7B-Instruct) across five benchmarks (MMLU, ARC-C, GSM8K, GPQA, HumanEval) and comparing against prior acceleration baselines (dLLM-Cache, dKV-Cache, Fast-dLLM). Reviewers report speedups on the order of ~3× on average, with specific headline throughput such as 37.29 TPS on GSM8K and a maximum reported 4.48× speedup, while maintaining near-baseline accuracy (often summarized as ≤1% difference in the reviews).

The main decision-relevant concerns raised across reviews focus on: (i) unquantified overhead of DCU/APD computations (e.g., token/layer metrics, cosine similarities, sorting, instability computation), (ii) hyperparameter robustness/adaptability, especially for longer generation lengths (>1k tokens) and task variation, (iii) long-form quality discussion, and (iv) incomplete baseline comparisons to certain cross-paradigm or industrial systems. In the rebuttal, the authors add multiple new measurements and analyses: a detailed wall-clock latency breakdown for computing token variation and layer-wise metrics and for DCU/APD overhead; longer-length experiments up to 1024 generated tokens on GSM8K; additional hyperparameter ablations for APD parameters (α, β) and budget parameters; an auto-tuning strategy scaling key budgets with generation length; analysis of Mandatory Update Window failure rate (token outside window); evaluation of threshold reset for APD on HumanEval; and additional testing on RTX 4090 for GSM8K and HumanEval. Two reviewers explicitly state their questions/concerns were addressed and that they keep their positive score.

**Reviewer Concerns:**

### Concerns addressed by the rebuttal / revision

* **Overhead quantification (m7U5, Pz1R, qCPz):**
  The authors provide latency breakdowns on GSM8K across output lengths (256/512/1024). For example, they report baseline inference time vs. DCU/APD inference times and “extra time” attributable to each mechanism (e.g., DCU Extra Time and APD Extra Time). They additionally break down token-variation time and layer-metric time (e.g., for 256 tokens: 0.65s and 1.02s vs. 29.16s baseline total time), and state overhead scales approximately linearly with output length.

* **Longer sequence validation and parameter scaling (>1k tokens) (m7U5, sidX, qCPz):**
  The rebuttal includes experiments with 1024 generated tokens on GSM8K. Results show that default settings (e.g., (B_{layer}=32, B_{window}=32)) can reduce performance at 1024 tokens (e.g., 73.92 in Table 3), while increasing budgets (e.g., 128/128) recovers performance (e.g., 79.15). The authors propose an auto-tuning strategy scaling the budgets with generation length (expressed as (B_{layer}=B_{window}=L/8), where (L) denotes generation length).

* **APD hyperparameter sensitivity and robustness (Pz1R, sidX):**
  The authors provide ablations over α and β (reported with both accuracy “Score” and “NFE”) and state performance is stable within the same order of magnitude, with best performance around α=0.001 and β=0.0008. They also provide results across generation lengths with fixed α/β (256/512/1024), including Dynamic-dLLM vs baseline scores and TPS.

* **Mandatory Update Window behavior and failure rate (Pz1R):**
  The authors report a failure rate definition (“token outside window”) and provide measured rates across generation lengths on GSM8K: 0.000 (256), 0.001 (512), 0.000 (1024). They justify the design via the claimed spatial locality of dLLM token dynamics.

* **Parallel decoding length and context length impact (m7U5):**
  The authors state Figure 6(c) shows that with fixed output length, fewer inference steps imply larger average parallel decoding length per step, with lower accuracy and faster inference speed; they also state their parameter configuration adapts to different context lengths, referencing results in a table (same Dynamic-dLLM vs baseline table across lengths).

* **Low-resource hardware practicality (sidX):**
  The authors add results on RTX 4090 for GSM8K and HumanEval, reporting baseline vs DCU vs Dynamic-dLLM scores and TPS (e.g., GSM8K baseline 78.34 / 5.48 TPS vs Dynamic-dLLM 77.91 / 29.77 TPS; HumanEval baseline 37.47 / 9.74 TPS vs Dynamic-dLLM 37.15 / 35.19 TPS).

* **Long-form / length-based quality discussion (m7U5):**
  The authors add evaluations across generation lengths (256/512/1024) showing Dynamic-dLLM scores close to baseline while TPS increases substantially, and they state this indicates no meaningful quality trade-off as length increases (within the tested settings).

* **Reviewer follow-up confirmations (m7U5, sidX):**
  Reviewer m7U5 states that the additional results addressed their questions. Reviewer sidX states they will keep their positive score.

### Concerns partially addressed:

* **Comparisons to cross-paradigm methods and industrial-grade dLLMs (sidX):**
  The authors do not provide direct comparisons to ES-dLLM or Mercury, citing constraints: ES-dLLM is a concurrent submission with no public preprint/implementation and uses hardware (H200) not accessible to them; Mercury is closed-source with no access to weights/implementation.

* **Deep dLLMs overhead at very high layer counts (qCPz):**
  Reviewer qCPz raised concerns about deep models (e.g., 128 layers). The authors report measurements on 32-layer models and state overhead scales approximately linearly with the number of layers, and note dLLM-Cache also scales linearly with layer count; they do not report direct experiments on 128-layer dLLMs.

* **Token sampling to reduce metric computation (qCPz):**
  The authors report experiments reducing token sample rate for variation checks (100% vs 50% vs 30% vs 0%), showing large accuracy drops at lower sample rates (e.g., 50% sample rate accuracy 40.21), and conclude sampling is not practically beneficial.

* **APD threshold reset strategy (qCPz):**
  The authors provide HumanEval experiments varying reset intervals (no reset vs 16/32/64 steps), showing similar accuracy with slightly increased inference steps under resets, and conclude resetting yields no meaningful gain and introduces overhead.

**Reviewer Scores:**

* **Reviewer m7U5:** Rating **6** (confidence 2). Post-rebuttal comment: “My questions have be addressed.”
* **Reviewer sidX:** Rating **6** (confidence 3). Post-rebuttal comment: “I would like to keep my positive score.”
* **Reviewer Pz1R:** Rating **6** (confidence 3). No post-rebuttal feedback provided, but the rebuttal claims added overhead breakdowns, ablations, and window failure-rate analysis addressing this reviewer’s listed questions and weaknesses.
* **Reviewer qCPz:** Rating **6** (confidence 3). No post-rebuttal feedback provided, but the rebuttal adds 1024-token experiments, overhead breakdowns, token sampling analysis, and APD threshold reset analysis aligned with the reviewer’s questions and weaknesses.

---

### Decision · Program_Chairs · 2026-01-26

Accept (Poster)